Imperial-TP-2025-CH-3

# Non-invertible symmetries of two-dimensional Non-Linear Sigma Models

**Guillermo Arias-Tamargo, Chris Hull, Maxwell L. Velásquez Cotini Hutt**

*Abdus Salam Centre for Theoretical Physics, The Blackett Laboratory, Imperial College London, Prince Consort Road, London, SW7 2AZ, UK*

*E-mail:* g.arias-tamargo@imperial.ac.uk, c.hull@imperial.ac.uk, m.hutt22@imperial.ac.uk

ABSTRACT: Global symmetries can be generalised to transformations generated by topological operators, including cases in which the topological operator does not have an inverse. A family of such topological operators are intimately related to dualities via the procedure of half-space gauging. In this work we discuss the construction of non-invertible defects based on T-duality in two dimensions, generalising the well-known case of the free compact boson to any Non-Linear Sigma Model with Wess-Zumino term which is T-dualisable. This requires that the target space has an isometry with compact orbits that acts without fixed points. Our approach allows us to include target spaces without non-trivial 1-cycles, does not require the NLSM to be conformal, and when it is conformal it does not need to be rational; moreover, it highlights the microscopic origin of the topological terms that are responsible for the non-invertibility of the defect. An interesting class of examples are Wess-Zumino-Witten models, which are self-dual under a discrete gauging of a subgroup of the isometry symmetry and so host a topological defect line with Tambara-Yamagami fusion. Along the way, we discuss how the usual 0-form symmetries match across T-dual models in target spaces without 1-cycles, and how global obstructions can prevent locally conserved currents from giving rise to topological operators.

# 1 Introduction

Symmetries [1] have been generalised [2] in various ways, by shifting the focus from explicit transformations of the fields to topological operators or defects in the quantum field theory (QFT). From this viewpoint, the topological operator generating a continuous global 0-form symmetry is the integral over a codimension-1 manifold of the usual Noether current. The action of the symmetry on a local operator is given by taking the topological operator for a surface surrounding it, then shrinking the surface so that the effect can be calculated using the OPE of the Noether current with the operator. The composition of transformations is then achieved by stacking several topological operators together. This generalisation has allowed for a deeper understanding of QFT, both by making new predictions about systems where generalised symmetries are present, as well as, equally importantly, by systematising and giving a common origin to various phenomena. The body of literature exploring this topic over the last decade is extensive; we recommend the following reviews [3–13].

There are various generalisations of symmetry (higher-form, higher-group, etc); we will focus here on the so-called *non-invertible* symmetries. These are generated by topological defects that do not necessarily have inverses, and whose fusion does not need to satisfy a group law. Determining what is the appropriate mathematical framework to understand these symmetries in full generality remains an ongoing enterprise (see the aforementioned reviews). What is clear is that they are present in many examples, with systematic ways of building non-invertible symmetries including discrete gaugings, higher gaugings (namely gaugings of higher-form symmetries in higher-codimension submanifolds), and half-space gaugings. This work focuses on the latter.

The idea behind the construction of non-invertible defects via half-space gauging is as follows [14, 15]. We begin with a theory $\mathcal{T}$ with an anomaly-free global symmetry $G$, which we take to be a finite group. We then split the space $W$ where $\mathcal{T}$ lives into two parts, $\Gamma_{\pm}$, with $\gamma$ the boundary between the two, and we gauge $G$ in $\Gamma_{+}$. This introduces new gauge fields $c$, so the resultant theory $\mathcal{T}/G$ on $\Gamma_{+}$ is in principle different from the original one. In certain situations, it can happen that $\mathcal{T}$ and $\mathcal{T}/G$ turn out to be the same theory, often thanks to a non-trivial duality. In this situation, the interface $\gamma$ becomes a defect living in a single theory. The situation is summarized in Figure 1. If we impose Dirichlet boundary conditions for the gauge fields,

$$c|_{\gamma} = 0 \,, \tag{1.1}$$

then this defect will be topological, because the change in the partition function when displacing $\gamma$ will be proportional to $dc$, which vanishes since $G$ is finite, and so $c$ is flat [14]. Due to the pivotal role that dualities play in these constructions, they are often referred to as *self-duality* defects or symmetries. Examples of models

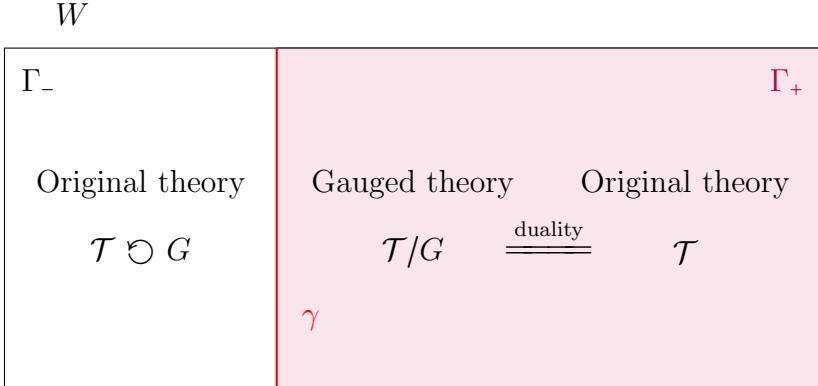

**Figure 1**. Idea of the construction of a half-space gauging or self-duality defect. We divide $W$ into two regions $\Gamma_-$ and $\Gamma_+$. If the original theory $\mathcal{T}$ has a global symmetry $G$, we can gauge it in $\Gamma_+$, and impose boundary conditions on the gauge fields such that the interface $\gamma$ is topological. If the gauged theory $\mathcal{T}/G$ is then dual to the original theory $\mathcal{T}$, the construction defines a topological operator in $\mathcal{T}$.

with these symmetries range from spin chains, where Kramers-Wannier (low-high temperature) dualities can be used to construct non-invertible symmetries at the critical temperature, to $\mathcal{N} = 4$ super Yang-Mills, which at the self-dual coupling $\tau = i$ also hosts such a symmetry as a consequence of S-duality [14, 15].

In two dimensions, the situation is well understood. From the formal point of view, the topological operators are either lines or points, and so a lot of the intricacies in higher dimensions, where topological operators of different dimensions can intersect in complicated ways giving rise to higher structures (see e.g. [16–20]), are absent. In two dimensions, the appropriate formalism is that of fusion categories [21, 22][1]. From a field-theoretic point of view, there are of course many simplifications that take place in low dimension, especially in the context of Conformal Field Theory (CFT). Indeed, the study of topological defect lines in 2d CFT has a long history, predating the advent of generalised symmetries [24–35].

One of the key examples of a half-space gauging (or self-duality) defect in 2d CFT is that of the free compact boson, or several copies thereof [33–41]. The model has a collection of U(1) symmetries with conserved currents proportional to $d\phi$ and $\star d\phi$ (where $\phi$ is the periodic scalar). The corresponding conserved quantities are the momentum and winding along the circle respectively. Gauging an anomaly-free $\mathbb{Z}_p$ subgroup of the momentum symmetry in half of the worldsheet puts us in the situation of Figure 1. Here, the relevant duality is T-duality, thanks to which the theory on $\Gamma_+$ after the $\mathbb{Z}_p$ gauging is the same as the theory on $\Gamma_-$ provided that the

---

[1]It has very recently been observed that this may not be the case, in the context of topological defects in the worldsheet of topological string theory [23].

radius is

$$R = \sqrt{\frac{p}{2\pi}}. \tag{1.2}$$

When this constraint is satisfied, the half-space gauging procedure gives rise to a non-invertible topological operator on $\gamma$ within the compact boson CFT.

The key observation is this: there is nothing particularly special about the torus as far as the derivation of T-duality is concerned. In fact, in the context of String Theory it is well known that the duality applies to very general Non-Linear Sigma Models (NLSMs) with Wess-Zumino (WZ) term [42–46]. One important part of the construction involves the gauging of global isometry symmetries of these models (see e.g. [47–51]) and another is how to account for flat but topologically non-trivial configurations of the gauge fields [45, 46, 52–54] (which indeed is required for the derivation of T-duality itself). Therefore, we have all the ingredients to ask the question: are there self-duality non-invertible symmetries in Non-Linear Sigma Models with WZ term?

The answer is that yes, there are indeed such non-invertible symmetries in self-dual NLSMs. The goal of this paper is to explicitly build the field configurations corresponding to these defects via the procedure of half-space gauging.[2] A NLSM whose target space has a U(1) isometry generated by a Killing vector field with compact orbit has a global U(1) *isometry symmetry*. The gauging of the full U(1) is possible provided certain obstructions are absent [47, 48] and we will be interested in the gauging of a $\mathbb{Z}_p$ subgroup of the U(1). We will restrict ourselves to the case in which the isometry is freely acting, i.e. without fixed points (this is a technical requirement whose origin will become clear throughout the paper). These are all the constraints which must be imposed on the sigma model in order to proceed with the half-space gauging. In particular, in our approach to the problem — namely, explicitly integrating out the discrete gauge fields at the level of the action — conformal symmetry will not play any role.

Our construction is also very general as far as the worldsheet $W$ is concerned. As we will see, we need it to be orientable (so that we can define the WZ term) and closed (which is another technical requirement that will become clear later). Thus, we can take $W$ to be any Riemann surface. The half-space gauging procedure of Figure 1 requires that the defect splits the worldsheet $W$ in two separate parts. This is not the case, for example, for a defect wrapping the $A$ or $B$-cycles of a torus. In particular, since $\gamma$ has to be the boundary of a half-worldsheet, it needs to be trivial in homology. As an example, a possible choice of $\gamma$ on a worldsheet of genus 2 is shown in Figure 2.

An important part of the story, which we discuss at length, is that we want to gauge discrete subgroups of the isometry symmetries in $\Gamma_+$, which is not the full

---

[2]For discussions of other non-invertible symmetries in NLSMs in dimensions higher than two, see [55–60].

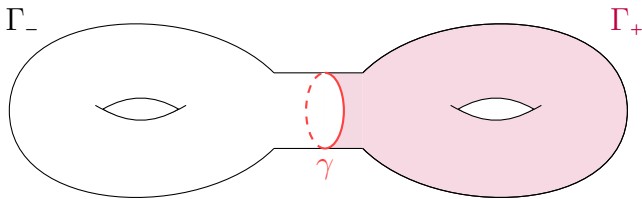

**Figure 2**. A valid choice of defect locus $\gamma$ for a worldsheet of genus 2. The defect $\gamma$ must be chosen to lie on a codimension-1 cycle which splits the worldsheet into two parts, $\Gamma_+$ and $\Gamma_-$.

worldsheet and, crucially, has a boundary. This gives rise to several subtleties which, at first glance, seem to hinder the objective of building a topological defect at the interface between the two halves of the worldsheet, $\gamma$. Firstly, in order for the WZ term to remain gauge-invariant on a worldsheet with boundary, it is necessary to introduce target space 1-form gauge fields. These can potentially yield new 't Hooft anomalies and new couplings for the worldsheet gauge fields $c$ [61]. Moreover, the boundary conditions which must be specified for a NLSM with boundary are generically modified by the gauging. These potential issues all have satisfying resolutions and we will show that the final result of the gauging in $\Gamma_+$ is the expected standard gauging in the bulk of $\Gamma_+$ plus a topological term localised on the boundary $\gamma$ of $\Gamma_+$. This topological term plays a crucial role for us, and we will see that it is this TQFT living on the defect which gives rise to the non-invertible fusion rules.

Following the logic summarised in Figure 1, once the discrete gauging is performed on half-space, we must identify the self-duality conditions that must be imposed so that the gauged theory on $\Gamma_+$ is the same as the original one on $\Gamma_-$. These conditions generalise the constraint (1.2) on the radius of the compact boson in order for the non-invertible defect to be present. In general, the self-duality constraints will involve other moduli, relating the strength of the $b$-field and the length of the orbits of the Killing vectors to the integer $p$. The final result is given by equations (4.44)–(4.45). This generalises to the case in which there is a freely-acting $U(1)^d$ isometry generated by $d$ commuting Killing vectors, in which case we will consider the gauging of a $\prod_m \mathbb{Z}_{p_{(m)}}$ subgroup; the self-duality conditions in this case appear in equation (4.49).

The rest of the paper is organised as follows. We begin in section 2 by reviewing the construction of the self-duality defects for the compact boson. Our aim is to show in this simple example that the discrete gauging proceeds in a very similar way to the derivation of T-duality, and how the TQFT on the defect arises from half-space gauging. With that in mind, we continue in section 3 by reviewing the derivation of T-duality for general NLSMs with WZ term; our emphasis lies in phrasing the calculation in a geometrical language where dealing with global aspects and discrete

gauge fields becomes transparent. We have included a discussion of the matching of global symmetries across T-dual models in the language of topological operators in subsection 3.4. This is non-trivial in cases where the target space does not have non-contractible 1-cycles, and while it is well-known to experts, the issues become transparent in the modern language. After that, section 4 contains the main result of this work, namely the discrete gauging in half of the worldsheet and the identification of self-duality conditions which must be satisfied for the theory to host a duality defect. In section 5 we give a few explicit examples. Amongst them, a particularly interesting class are Wess-Zumino-Witten (WZW) models where, surprisingly, the self-duality condition coincides precisely with the quantisation condition required by conformal symmetry [62]. As a result, we explicitly find non-invertible defects for $\mathrm{SU}(N)_\kappa$ WZW models at all levels $\kappa$, which in general are not Verlinde lines. We conclude in section 6 with a discussion of some open questions. In appendix A we collect some useful results used in the main text and give our conventions and notation. Some of the more technical derivations have been relegated to appendix B.

## 2 Warm-up: the compact boson

In the discussion of duality defects giving rise to generalised symmetries, one of the simplest examples, examined in [33–35] (and revisited more recently in e.g. [36–41]), is that of a compact boson in two dimensions. In this section, we review their construction, establishing the notation and conventions that we will use in later sections.

The action is

$$S = \frac{R^2}{2} \int_W d\phi \wedge \star d\phi \,, \tag{2.1}$$

and the scalar field $\phi$ is periodic with period $2\pi$; which is to say, $\phi$ is a map $\phi : W \to S_R^1$. Here $W$ is a 2 dimensional worldsheet, and the symbol $\star$ denotes the Hodge star on $W$. We will present our analysis in Lorentzian signature, so that $\star^2 = +1$ when acting on worldsheet 1-forms, but much of our discussion is readily continued to Euclidean signature. Often we will write $d\phi \wedge \star d\phi = d\phi \cdot d\phi = d\phi^2$.

The action (2.1) is invariant under constant shifts $\phi \to \phi + h$, whose associated conserved charge is usually called momentum. It is instructive to understand it as a global symmetry related to the $\mathrm{U}(1)$ isometry of the target space $S_R^1$, which is generated by the Killing vector $\partial_\phi$. This model also has a winding symmetry, but this will not generalise to NLSMs whose target space has trivial fundamental group.

### 2.1 T-duality of the compact boson

The T-dual of the compact boson is derived by coupling a background gauge field for the isometry, and then adding a Lagrange multiplier term which enforces that

the background field is trivial; in this way, we ensure that we have not modified the path integral. This is achieved by promoting the derivative to a covariant derivative

$$d\phi \rightarrow D\phi = d\phi - C\,, \tag{2.2}$$

where $C$ is the 1-form gauge field, transforming as $C \rightarrow C + dh$, and adding the Lagrange multiplier term[3]

$$\frac{1}{2\pi} \int C \wedge d\widetilde{\phi}. \tag{2.3}$$

where $\widetilde{\phi}$ is another $2\pi$-periodic scalar. The action is then

$$\hat{S}[\phi, \widetilde{\phi}, C] = \frac{R^2}{2} \int_W (d\phi - C) \wedge \star (d\phi - C) + \frac{1}{2\pi} \int_W C \wedge d\widetilde{\phi}. \tag{2.4}$$

As usual, the quantisation involves continuing to Euclidean signature and performing the Euclidean path integral, and then continuing back. However, it is also interesting to consider the Euclidean functional integral in its own right, allowing a Riemann surface that may have many 1-cycles.

If we first integrate over $\widetilde{\phi}$, then $\widetilde{\phi}$ acts as a Lagrange multiplier field and its equation of motion enforces that $dC = 0$. Moreover, since $\widetilde{\phi}$ is periodic with period $2\pi$, summing over the periods of $d\widetilde{\phi}$ forces $C$ to have trivial holonomy around all 1-cycles. That is, integrating out $\widetilde{\phi}$ implies that $C$ is a completely trivial gauge field. Then we can fix $C = 0$ via a choice of gauge, and we go back to the original action (2.1).

On the other hand, if we choose to perform the integral over $C$ first, we arrive at the T-dual model for $\widetilde{\phi}$. We begin by rewriting the action (2.4) as

$$\hat{S} = \frac{R^2}{2} \int_W \left[ C^2 - 2C \cdot \left( d\phi - \frac{1}{2\pi R^2} \star d\widetilde{\phi} \right) + d\phi^2 \right]. \tag{2.5}$$

Completing the square and shifting $C \rightarrow C'$,

$$\hat{S} = \frac{R^2}{2} \int_W \left[ C'^2 - \left( d\phi - \frac{1}{2\pi R^2} \star d\widetilde{\phi} \right)^2 + d\phi^2 \right]. \tag{2.6}$$

Now we can easily perform the path integral over $C'$, since the action is quadratic. It results in an overall normalization for the partition function, which will drop out of any correlator, and which we can safely ignore. The rest of the action results in

$$\hat{S} = \frac{1}{2\pi} \int_W d\phi \wedge d\widetilde{\phi} + \frac{1}{8\pi^2 R^2} \int_W d\widetilde{\phi} \wedge \star d\widetilde{\phi}. \tag{2.7}$$

---

[3]Let us remark again that we work in Lorentzian signature, such that single derivative terms differ by a factor of $i$ with respect to much of the recent literature in generalised symmetries, which is presented in Euclidean signature.

Since both $\phi$ and $\widetilde{\phi}$ are periodic with period $2\pi$, the first term contributes a factor $\exp(2\pi i \mathbb{Z})$ to the path integral and so can be neglected. Then doing the path integral over $\phi$ is trivial, and we obtain the dual action

$$\widetilde{S} = \frac{1}{8\pi^2 R^2} \int_W d\widetilde{\phi} \wedge \star d\widetilde{\phi}, \tag{2.8}$$

which is a Sigma Model with target space $S^1$ of radius $1/(2\pi R)$. This action also has an isometry symmetry generated by the (target space) Killing vector $\partial_{\widetilde{\phi}}$, which shifts $\widetilde{\phi}$ by a constant. In fact, we observe that this symmetry was already present in (2.4).

## 2.2  Discrete gauging for the compact boson

Next we discuss a gauging construction that is very similar to the T-duality discussed in the previous section. The idea is to couple the theory to a U(1) gauge field as before, but instead of imposing the condition that the gauge field is trivial, we now impose the condition that the gauge field is flat, but can have non-trivial $\mathbb{Z}_p$-valued holonomies around 1-cycles in $W$. The only difference to the T-duality construction is that the Lagrange multiplier term, which was previously of the form (2.3), is now modified by adding a factor of $p$.

We then have the gauged action with Lagrange multiplier term

$$\hat{S}_{(p)} = \frac{R^2}{2} \int_W (d\phi - c) \wedge \star (d\phi - c) + \frac{p}{2\pi} \int_W c \wedge d\widetilde{\phi}. \tag{2.9}$$

We are now switching notation for the gauge field from $C$ to $c$, to emphasise that there is a difference with respect to $C$. The equation of motion of the Lagrange multiplier field $\widetilde{\phi}$ here implies that the holonomies of $c$ are $\mathbb{Z}_p$-valued, and so the gauge field $c$ is allowed to have topologically non-trivial configurations. In the previous discussion of T-duality, with gauged action (2.4), integrating out $\widetilde{\phi}$ implied that $C$ was flat and had vanishing holonomies, and so was pure gauge. We emphasise that this gauging operation is not a duality and the resulting theory will generically differ from the original one.

As before, we proceed with the Gaussian path integral over $c$ by rewriting the action as

$$\hat{S}_{(p)} = \frac{R^2}{2} \int_W \left[ c^2 - 2c \cdot \left( d\phi - \frac{p}{2\pi R^2} \star d\widetilde{\phi} \right) + d\phi^2 \right], \tag{2.10}$$

and after doing the integrals in $c$ and $\phi$ we find

$$S'_{(p)} = \frac{p^2}{8\pi^2 R^2} \int_W d\widetilde{\phi} \wedge \star d\widetilde{\phi}. \tag{2.11}$$

up to a boundary term which will be discussed shortly in section 2.3.

We conclude that integrating out the $\mathbb{Z}_p$ gauge field leads to a NLSM whose target space is a circle of radius $R' = p/(2\pi R)$. Thus, there is a self-dual radius at which $R' = R$ so that the action (2.11) is the same as the original action (2.1) (up to an important boundary term). In our conventions, where both $\phi$ and $\widetilde{\phi}$ are $2\pi$-periodic, this radius is

$$R = \sqrt{\frac{p}{2\pi}}\,. \tag{2.12}$$

Let us finish this section with a brief comment. There is a $\mathbb{Z}_p$ subgroup of the shift symmetry under which $\phi \to \phi + 2\pi/p$. Gauging this $\mathbb{Z}_p$ is an orbifolding that changes the radius from $R$ to $R/p$, and a subsequent T-duality then takes this to a compact boson with radius $p/(2\pi R)$. On the other hand, adding a coupling to a flat gauge field with $\mathbb{Z}_p$ holonomy led us directly to a radius $p/(2\pi R)$. That is, these path integral manipulations are equivalent to orbifolding by a $\mathbb{Z}_p$ subgroup and subsequently performing a T-duality. Of course, T-dual theories are physically equivalent, so that the theory constructed in this way with radius $p/(2\pi R)$ is physically equivalent to the $\mathbb{Z}_p$ gauged theory. Note that for $p = 1$ the construction is simply a T-duality.

## 2.3 Self-duality defects of the compact boson

Self-duality non-invertible defects for the compact boson can be defined using the general construction of half-space gauging. In half of the worldsheet (say, for coordinate $\sigma^1 < 0$) we have our original theory. In the other half of the worldsheet ($\sigma^1 > 0$) we gauge a discrete subgroup of a global symmetry. If we impose Dirichlet boundary conditions for the discrete gauge field at the boundary $\sigma^1 = 0$, we generically end with a topological interface between two different theories (see Figure 1). In certain cases, we can tune parameters such that the gauged theory is the same as the original one, in which case the interface defines a topological defect sitting in a single theory. Generically, this construction leads to non-invertible defects [14, 15].

In the example at hand, the half-space gauging is taken to be the construction of the previous subsection, which is a T-dual formulation of a $\mathbb{Z}_p$ gauging. We divide the worldsheet $W$ in two regions, $\Gamma_-$ and $\Gamma_+$, separated by a boundary $\gamma$. In $\Gamma_-$, the theory is the compact boson of radius $R$ defined by the action (2.1). In $\Gamma_+$, we have the theory with action (2.9) which then leads to the action (2.11). At the self-dual radius given by (2.12) the theory in $\Gamma_+$ is the same as the one in $\Gamma_-$.

In this situation, we define a topological defect by the action (2.1) in $\Gamma_-$ and the action (2.11) in $\Gamma_+ \cup \gamma$, together with the topological boundary condition $c|_\gamma = 0$ to ensure continuity. It has been shown in [14, 15] that these defects are guaranteed to satisfy the fusion rules of a Tambara-Yamagami category $\mathrm{TY}(\mathbb{Z}_p)$. In particular, if the topological line operator generating the $\mathbb{Z}_p$ subgroup of the isometry is denoted by $\eta(\sigma)$, where $\sigma$ is a 1-cycle, and the non-invertible defect defined by performing

the half-space gauging procedure is denoted by $\mathcal{N}$, then the fusion rules are

$$\eta^p = 1, \quad \eta\mathcal{N} = \mathcal{N}\eta, \quad \mathcal{N} \times \overline{\mathcal{N}} = \sum_{\sigma \in H_1(\gamma, \mathbb{Z}_p)} \eta(\sigma). \tag{2.13}$$

Commonly, the non-invertible fusion which a given defect exhibits is a result of a topological theory living on the defect, which may interact also with the bulk fields. In the example of the compact boson, we can understand this topological theory from the half-space gauging procedure described in subsection 2.2. In particular, when going from (2.10) to (2.11), a term analogous to the first term in (2.7) also arises, giving

$$\frac{p}{2\pi} \int_{\Gamma_+} d\phi \wedge d\widetilde{\phi}. \tag{2.14}$$

This term is a total derivative and can, therefore, be written as an integral on $\partial\Gamma_+ = \gamma$, which is precisely the location of the defect. The resulting TQFT is a 1-dimensional BF-type theory,

$$S_\gamma = \frac{p}{2\pi} \int_\gamma \phi\, d\widetilde{\phi}. \tag{2.15}$$

Note that the fields $\phi$ and $\widetilde{\phi}$ are not globally well defined coordinates and therefore it is not rigorous to apply Stokes' theorem to this situation. Nevertheless, (2.15) still gives a well-defined contribution to the path integral when $\phi \sim \phi + 2\pi$.

An independent way to check this claim was discussed in [40]. Observe that in the gauged theory (2.9), before integrating out $c$ and $\phi$, the equations of motion for $c$ are

$$c = d\phi - \frac{p}{2\pi R^2} \star d\widetilde{\phi}. \tag{2.16}$$

On the locus of the defect the gauge field satisfies the boundary condition $c|_\gamma = 0$, which allows us to relate the fields on the left and on the right,

$$d\phi\Big|_\gamma = \frac{p}{2\pi R^2} \star d\widetilde{\phi}\Big|_\gamma = \star d\widetilde{\phi}\Big|_\gamma, \tag{2.17}$$

where this equation is meant to be satisfied only on-shell, and in the second equality we are imposing the value of the self-dual radius (2.12). We can then ask which action $S_\gamma$ should describe the TQFT on the defect such that (2.17) arises as an equation of motion. That is, we ask which defect must be inserted such that its presence has precisely the same effect on the theory as gauging a $\mathbb{Z}_p$ subgroup of the U(1) isometry symmetry in $\Gamma_+$.

Let us momentarily assume that we do not know the TQFT (2.15) on the defect, and consider a general topological action $S_\gamma = \int_\gamma \mathcal{L}_\gamma$. The total action is then

$$S = S_- + S_\gamma + S_+ \tag{2.18}$$

$$= \frac{p}{4\pi} \int_{\Gamma_-} d\phi \wedge \star d\phi + \int_\gamma \mathcal{L}_\gamma + \frac{p}{4\pi} \int_{\Gamma_+} d\widetilde{\phi} \wedge \star d\widetilde{\phi} \tag{2.19}$$

In general, the defect Lagrangian $\mathcal{L}_\gamma$ can depend on the bulk fields $\phi$ and $\widetilde\phi$ as well as other degrees of freedom localised to the defect. Let us make the simplifying assumption that there are no such fields living only on $\gamma$; that is, $S_\gamma$ is a functional of only $\phi$ and $\widetilde\phi$. The variation of the action with respect to $\phi$ and $\widetilde\phi$ gives rise to a bulk and a boundary contribution in the equations of motion. The former is the expected equation of motion for the free scalar $\phi$ in $\Gamma_-$ or $\widetilde\phi$ in $\Gamma_+$. We are interested in the contributions on the boundary,

$$\frac{p}{2\pi} \star d\phi|_\gamma - \left.\frac{\partial \mathcal{L}_\gamma}{\partial \phi}\right|_\gamma + d\left(\left.\frac{\mathcal{L}_\gamma}{\partial(d\phi)}\right)\right|_\gamma = 0 \,, \tag{2.20}$$

$$\frac{p}{2\pi} \star d\widetilde\phi|_\gamma + \left.\frac{\partial \mathcal{L}_\gamma}{\partial \widetilde\phi}\right|_\gamma - d\left(\left.\frac{\mathcal{L}_\gamma}{\partial(d\widetilde\phi)}\right)\right|_\gamma = 0 \,. \tag{2.21}$$

The goal is to have (2.17) arise from these equations of motion. It is straightforward to verify that this is achieved precisely if the action on the defect is given by (2.15). This topological theory on the defect also leads directly to the non-invertible fusion (2.13) of the duality defect, as explained in detail in the appendix of [40].

## 3 Recap: T-duality from the gauged NLSM

The main objective of this paper is to generalise the above construction of the non-invertible defect of the compact boson to a more general class of NLSMs. Given that the procedure to define the non-invertible defect is very similar to the derivation of the T-duality, our goal in this section is to review how the latter is derived in the context of NLSMs, mostly following [46]. Indeed, in section 4 we will find that only a minor modification is needed in order to find non-invertible symmetries in NLSMs.

This section is organised as follows. We begin in subsection 3.1 by discussing the isometry symmetry of NLSMs with WZ term and how to gauge it. There is a potential obstruction to the gauging that can be viewed as a 't Hooft anomaly for the symmetry, but when we add Lagrange multipliers imposing that the gauging is trivial the obstruction to the gauging is weaker; we review this construction in subsection 3.2. We proceed to derive the T-dual NLSM in subsection 3.3 by integrating out the gauge fields. For T-duality on a product space $M = N \times T^d$, the duality exchanges momentum and winding modes. As we will review, the duality is present for any target space $M$ with a compact isometry acting without fixed points, even when $M$ does not have non-trivial 1-cycles. Without 1-cycles there are no winding modes so that the T-duality is not as simple as momentum and winding being exchanged. In subsection 3.4 we discuss this issue in the modern language of topological operators. We conclude in subsection 3.5 with the detailed example of a NLSM with target space $S^3$, which is T-dual to a NLSM with target space $S^2 \times S^1$ with non-trivial $H$-flux. Our conventions and some of our terminology are given in appendix A.

### 3.1 The Gauged Non-Linear Sigma Model

The starting point is the following NLSM, which is a theory of maps $\Phi : W \to M$ with a WZ term,

$$S = \frac{1}{2} \int_W g_{ij} \, dX^i \wedge \star dX^j + \frac{1}{2} \int_W b_{ij} \, dX^i \wedge dX^j \, . \tag{3.1}$$

where $X^i$ denote coordinates on the target space $M$, so the maps $\Phi$ are given locally by functions $X^i(\sigma)$ where $\sigma^a$ are the coordinates on the worldsheet $W$. While $b$ locally defines a 2-form on $M$, globally it should be understood as a gerbe connection over $M$. The definition of the WZ term in (3.1) therefore needs clarification. It will be useful to write this term instead as an integral on a three dimensional manifold $V$ such that $W = \partial V$,[4]

$$S = \frac{1}{2} \int_W g_{ij} \, dX^i \wedge \star dX^j + \frac{1}{3} \int_V H_{ijk} \, dX^i \wedge dX^j \wedge dX^k \, , \tag{3.2}$$

where $H = \mathrm{d}b$ (see appendix A for our conventions for differential forms). A definition of the WZ term purely in terms of $b$ is also possible using a generalisation of the Wu-Yang procedure [63].

Suppose that the target space $M$ has a collection of commuting $\mathrm{U}(1)$ isometries generated by a set of Killing vectors $k_m$, with $m = 1, \ldots, d$, such that

$$\mathcal{L}_m g = 0, \quad \mathcal{L}_m H = 0 \, , \tag{3.3}$$

where $\mathcal{L}_m \equiv \mathcal{L}_{k_m}$ denotes the Lie derivative with respect to the vector field $k_m$. If the isometries act without fixed points, then $M$ is a $T^d$ bundle over a base space $N$,

$$T^d \hookrightarrow M \twoheadrightarrow N \, , \tag{3.4}$$

and its metric can be written

$$g = \bar{g} + G_{mn} \xi^m \otimes \xi^n \tag{3.5}$$

where $\bar{g}$ is a metric on $N$, $G_{mn}$ is

$$G_{mn} = g_{ij} k_m^i k_n^j \, , \tag{3.6}$$

and $\xi^m$ is the 1-form dual to $k^m$, such that $\xi^m(k_n) = \delta_n^m$. The $\xi^m$ have components

$$\xi_i^m = G^{mn} g_{ij} k_n^j \, , \tag{3.7}$$

---

[4]The path integral does not depend on the choice of $V$ provided the flux is appropriately quantised such that $\frac{1}{2\pi}[H]$ defines an integral cohomology class.

where $G^{mn}$ are components of the inverse matrix of $G_{mn}$ defined in (3.6).[5] We will denote the coordinates on $M$ by $X^i = (Y^\mu, X^m)$ where $Y^\mu$ are coordinates on the base $N$, and $X^m$ are coordinates on the $T^d$ fibres, adapted such that

$$k_m = \frac{\partial}{\partial X^m}\,. \tag{3.8}$$

The transformation

$$\delta X^i = \alpha^m k_m^i\,, \tag{3.9}$$

where $\alpha^m$ are constants, is a global symmetry of the NLSM (3.2) on $M$ if the contraction $\iota_m H \equiv \iota_{k_m} H$ is exact [47] (note that it follows from (3.3) that $\iota_m H$ is closed); that is, if there exist globally-defined 1-forms $v_m$ on $M$ such that

$$\iota_m H = \frac{1}{2\pi} \mathrm{d} v_m\,. \tag{3.10}$$

Note that there is a freedom to shift $v_m$ by an arbitrary closed 1-form.

T-duality is derived, as for the compact boson, by 1) gauging this isometry and 2) imposing that the gauge fields are trivial with a Lagrange multiplier field. The gauging involves adding terms to the action involving 1-form gauge fields $C^m$ on $W$ so that the action is invariant under isometry transformations in which the parameters are local functions $\alpha^m(\sigma)$. In the absence of a WZ term, this involves replacing the partial derivatives by suitable covariant derivatives, but as the WZ term changes by a total derivative under rigid isometries (with constant $\alpha$), the gauging of this term is non-trivial in general. There is a potential obstruction to this gauging [47] that can be viewed as a 't Hooft anomaly for the rigid symmetry. It was shown in [47] that this anomaly is absent provided that the $v_m$ in (3.10) can be chosen such that the following conditions are met:

$$\mathcal{L}_m v_n = 0, \quad \iota_m v_n + \iota_n v_m = 0\,. \tag{3.11}$$

In this case, gauge fields $C^m$ can be introduced on $W$ transforming as

$$C^m \to C^m + d\alpha^m \tag{3.12}$$

to gauge the NLSM on $M$. The gauged NLSM (GNLSM) action can be written in the manifestly gauge-invariant form [47]

$$S_{\text{gauged}} = \frac{1}{2} \int_W g_{ij} DX^i \wedge \star DX^j + \int_V \left( \frac{1}{3} H_{ijk} DX^i \wedge DX^j \wedge DX^k + \frac{1}{2\pi} dC^m \wedge v_{mi} DX^i \right), \tag{3.13}$$

---

[5]As the isometry group acts without fixed points the Killing vectors $k^m$ are nowhere vanishing and so $G_{mn}$ is invertible.

where the covariant derivative $DX^i$ is defined by

$$DX^i = dX^i - C^m k_m^i \,. \tag{3.14}$$

This action can in fact be written even for models with a 't Hooft anomaly; the conditions (3.11) arise from requiring the 3-dimensional action give 2-dimensional field equations on the boundary $W$ of $V$ [47]. In the case where the anomaly does not vanish, the action (3.13) can be interpreted as containing a Chern-Simons anomaly inflow term (given in [47]) which depends on the choice of $V$. The anomaly can be understood as this obstruction to writing the GNLSM as a theory on $W$.

In fact, a more general construction is possible even when the restrictive condition (3.10) is not met; that is, when $\iota_m H$ is closed but not exact [46]. The idea is to understand $\iota_m H$ as the curvature for an auxiliary $U(1)^d$ fibre bundle over spacetime. This bundle is constructed in such a way that 1) the (lift of) the isometries can be gauged and 2) the extra fibre coordinates act as Lagrange multiplier fields imposing that the gauge fields $C^m$ are pure gauge. By making different gauge choices, one can then move between two different descriptions of the same quantum theory. One such gauge choice, where we set $C^m = 0$, returns us to the original NLSM (3.2), while another one takes us to a T-dual NLSM which can appear very different from the original theory. We refer to this construction as the doubled GNLSM, and we review it in the following subsection.

## 3.2 The doubled Gauged Non-Linear Sigma Model

Let $\{U_\alpha\}$ be open sets covering $M$. Since $\iota_m H$ is closed, it is locally exact in each open patch, so there exist non-globally defined 1-forms $v_m^\alpha$ such that

$$\iota_m H = \frac{1}{2\pi} \mathrm{d} v_m^\alpha \tag{3.15}$$

in $U_\alpha$. If the $v_m^\alpha$ obey a cocycle condition, they can be viewed as the components of a connection on a larger space $\hat{M}$ which is a $T^d$ bundle over $M$ [46]. However, $M$ is itself a $T^d$ bundle over a base $N$. In special cases in which the new $T^d$ fibres of $\hat{M}$ are trivially fibred over the $T^d$ fibres of $M$, $\hat{M}$ can be viewed as a $T^{2d}$ bundle over $N$,

$$T^{2d} \hookrightarrow \hat{M} \twoheadrightarrow N \tag{3.16}$$

with fibres the 'doubled torus' $T^{2d}$.

This bundle $\hat{M}$ over $M$ can itself be used as the target space of a NLSM, as we now review. Let us denote the coordinates on this second $T^d$ fibre by $\hat{X}_m^\alpha$ in each patch. Overall, the coordinates on $\hat{M}$ are $\hat{X}^I = (Y^\mu, X^m, \hat{X}_m)$ where $Y^\mu$ are coordinates on the base $N$, $X^m$ are coordinates on the $T^d$ fibres of $M$, and $\hat{X}_m$ are coordinates on the remaining $T^d$ fibres of $\hat{M}$. The transition functions of the $\hat{X}^m$ are chosen such that the 1-forms

$$\hat{v}_m = v_m^\alpha + \mathrm{d} \hat{X}_m^\alpha \tag{3.17}$$

are globally defined on $\hat{M}$. Furthermore, $g$ and $H$ can be pulled back (via the natural projection) to a metric $\hat{g}$ and closed 3-form $\hat{H}$ on $\hat{M}$ whose only non-zero components are

$$\hat{g}_{ij} = g_{ij}, \quad \hat{H}_{ijk} = H_{ijk}. \tag{3.18}$$

That is, the components of $\hat{g}$ and $\hat{H}$ along the $\hat{X}_m$ fibre directions all vanish. Therefore, defining a NLSM on $\hat{M}$ as in (3.2) with metric $\hat{g}$ and closed 3-form $\hat{H}$ actually has an action which is equal to that of the NLSM on $M$ given in (3.2) with metric $g$ and closed 3-form $H$. In other words, we can consider $\hat{M}$ as the target space of the NLSM, instead of $M$.

Even if the NLSM on $M$ does not satisfy the conditions (3.11), it is possible to satisfy them on $\hat{M}$ by demanding that

$$\hat{\mathcal{L}}_m \hat{v}_n = 0, \tag{3.19}$$

$$\hat{\imath}_m \hat{v}_n + \hat{\imath}_n \hat{v}_m = 0, \tag{3.20}$$

where $\hat{\imath}_m$ and $\hat{\mathcal{L}}_m$ denote contraction with and Lie derivative with respect to $\hat{k}_m$ respectively. Here, $\hat{k}_m$ is a lift of the Killing vector $k_m$ from $M$ to $\hat{M}$, which is of the form

$$\hat{k}_m = k_m + \Theta_{mn} \frac{\partial}{\partial \hat{X}_n} \tag{3.21}$$

for some $\Theta_{mn}(X^i)$. Since $\hat{g}$ and $\hat{H}$ are independent of $\hat{X}_m$, the $\hat{k}_m$ automatically satisfy

$$\hat{\mathcal{L}}_m \hat{g} = 0, \quad \hat{\mathcal{L}}_m \hat{H} = 0, \tag{3.22}$$

and so generate isometries of $\hat{M}$ for any $\Theta_{mn}$. Then (3.20) can be satisfied by choosing a $\Theta_{mn}$ such that

$$\hat{\imath}_m \hat{v}_n = \iota_m v_n + \Theta_{mn} = 2\pi B_{mn}, \tag{3.23}$$

for some antisymmetric $B_{mn} = -B_{nm}$. In fact, one can use the freedom to shift $v_m$ (3.10) by an exact 1-form to set $\Theta_{mn} = 0$ locally [64]. In order to avoid potential global issues with this redefinition, we will maintain $\Theta_{mn}$ throughout. Furthermore, from (3.17), (3.21), and (3.23) it follows that (3.19) is satisfied if

$$\iota_m \iota_n H = -\mathrm{d}B_{mn}. \tag{3.24}$$

We note that the form of $\hat{g}$ and $\hat{H}$ also imply that there are another $d$ commuting Killing vectors on $\hat{M}$,

$$\widetilde{k}^m = \frac{\partial}{\partial \hat{X}_m}. \tag{3.25}$$

These isometries generate the cycles around the second set of $T^d$ fibres in $\hat{M}$. We will see below that (loosely speaking) under T-duality the two sets of $T^d$ fibres in $\hat{M}$ are exchanged. In the simple case where $M = N \times T^d$ is a product space, the Killing

vectors (3.25) then give a geometric origin for the winding symmetries, placing them on equal footing with the isometry symmetries related to the $\hat{k}_m$ (which are commonly called 'momentum symmetries' in the torus case). More broadly, the $\widetilde{k}^m$ isometries generalise the winding symmetry of the torus example to target spaces that do not have non-trivial 1-cycles.

Let us consider the transformations of the NLSM on $\hat{M}$ generated by the $\hat{k}_m$ isometries,

$$\delta \hat{X}^I = \alpha^m \hat{k}_m^I \,, \tag{3.26}$$

for constants $\alpha^m$. As discussed above, in order for this to be a global symmetry of the NLSM on $\hat{M}$ we need that $\hat{\imath}_m \hat{H}$ is exact. From (3.21), we find

$$\hat{\imath}_m \hat{H} = \frac{1}{2\pi} \mathrm{d} \hat{v}_m \,. \tag{3.27}$$

Since $\hat{v}_m$ are globally defined, this implies that $\hat{\imath}_m \hat{H}$ is, indeed, exact and so (3.26) is a global symmetry which, from (3.19)–(3.20), does not have a 't Hooft anomaly. Moreover, by assumption we are restricting ourselves to the case where the isometries generated by the $\hat{k}_m$ commute, which is the case if[6]

$$\iota_m \iota_n \iota_p H = 0 \,. \tag{3.28}$$

The gauging of the global symmetry (3.26) of the NLSM on $\hat{M}$ can then be achieved in a manner analogous to (3.13) by introducing gauge fields $C^m$ transforming as in (3.12), and defining covariant derivatives

$$D\hat{X}^I = d\hat{X}^I - C^m \hat{k}_m^I \,. \tag{3.29}$$

We refer to the resulting gauged NLSM on $\hat{M}$ as the doubled GNLSM. Its full action is then the analogue of (3.13),

$$\hat{S} = \frac{1}{2} \int_W g_{ij} \, DX^i \wedge \star DX^j + \frac{1}{3} \int_V H_{ijk} \, DX^i \wedge DX^j \wedge DX^k + \frac{1}{2\pi} \int_V dC^m \wedge \hat{v}_{mI} D\hat{X}^I \,. \tag{3.30}$$

Note that the $\hat{X}_m$ coordinates do not appear in the first two terms as a consequence of (3.18). The final term in (3.30) includes a contribution of the form $\frac{1}{2\pi} \int_V dC^m \wedge d\hat{X}_m = \frac{1}{2\pi} \int_W C^m \wedge d\hat{X}_m$ such that the doubled fibre coordinates $\hat{X}_m$ act as the Lagrange multiplier fields which impose that the $C^m$ are pure gauge. It is this feature which implies that this construction does not change the partition function and, therefore, can be used to derive T-dual descriptions of the theory. As a sanity check, it is simple to see that (3.30) reduces to (2.4) in the case where the target space is $S^1$ and there is no $b$-field.

---

[6]More generally, one has that the commutator $[\hat{k}_m, \hat{k}_n] = -\left(\iota_m \iota_n \iota_p H\right) \partial_{\hat{X}_p}$ [46].

While the construction of the doubled GNLSM is subtle, the main insight of [46] is that it can be described using (globally defined) curvatures on the base space $N$. The 1-forms $\xi^m$ defined in (3.7) satisfy $\widetilde{\iota}_m \xi^n = 0$, where $\widetilde{\iota}_m \equiv \iota_{\widetilde{k}_m}$, and so can be written

$$\xi^m = A^m + \mathrm{d}X^m \tag{3.31}$$

where $A^m$ satisfy

$$\hat{\iota}_m A^n = 0, \quad \widetilde{\iota}_m A^n = 0, \quad \mathcal{L}_m A^n = 0, \quad \mathcal{L}_{\widetilde{m}} A^n = 0 \,. \tag{3.32}$$

Forms satisfying these conditions are referred to as basic forms (see appendix A). In other words, the $A^m$ can be seen as connection 1-forms defined on $N$, i.e. $A^m = A^m_\mu(Y)\mathrm{d}Y^\mu$, which describe $M$ as a $T^d$ bundle over $N$ as in (3.4). The curvature of this bundle is then

$$F^m = \mathrm{d}A^m = \mathrm{d}\xi^m \tag{3.33}$$

which also satisfies $\hat{\iota}_m F^n = 0$ and $\widetilde{\iota}_m F^n = 0$ and so is a globally defined 2-form on $N$. $\frac{1}{2\pi}[F^m]$ are the first Chern classes of the bundle (3.4).

With a similar objective, we define the 1-forms[7]

$$\widetilde{\xi}_m = \hat{v}_m + 2\pi B_{mn} \xi^n \tag{3.34}$$

which satisfy $\hat{\iota}_m \widetilde{\xi}^n = 0$. It is useful to introduce a change of coordinates $(Y^\mu, X^m, \hat{X}_m) \to (Y^\mu, X^m, \widetilde{X}_m)$, adapted such that the $2d$ Killing vectors $\hat{k}_m$, $\widetilde{k}^m$ take a particularly simple form

$$\hat{k}_m = \frac{\partial}{\partial X^m}, \quad \widetilde{k}^m = \frac{\partial}{\partial \widetilde{X}_m} \,. \tag{3.35}$$

This can be achieved with

$$\widetilde{X}_m = \hat{X}_m + \eta_m \,, \quad \text{with} \quad \frac{\partial \eta_m}{\partial X^n} = -\Theta_{mn} \,. \tag{3.36}$$

Then the 1-forms $\widetilde{\xi}_m$ can be written

$$\widetilde{\xi}_m = \widetilde{A}_m + \mathrm{d}\widetilde{X}_m \tag{3.37}$$

where the $\widetilde{A}_m$ satisfy $\hat{\iota}_m \widetilde{A}_n = 0$ and $\widetilde{\iota}_m \widetilde{A}_n = 0$, and so can also be seen as connection 1-forms on $N$, describing the doubled fibres of the $\hat{M}$ geometry. The curvature 2-form associated with $\widetilde{A}_m$ is

$$\widetilde{F}_m = \mathrm{d}\widetilde{A}_m = \mathrm{d}\widetilde{\xi}_m \tag{3.38}$$

which also satisfies $\hat{\iota}_m \widetilde{F}_n = 0$ and $\widetilde{\iota}_m \widetilde{F}_n = 0$, and so is a globally defined basic 2-form. The curvatures $\widetilde{F}_m$ encode the topology of the $T^d$ bundle over $M$ which was

---

[7]The normalisation is chosen for later convenience, such the periods of both $F^m$ and $\widetilde{F}_m$ have the same $2\pi\mathbb{Z}$ quantisation.

originally described by the connections $v_m$ in (3.15). Since the $v_m$ are related to $H$, the curvatures $\widetilde{F}_m$ are often referred to as $H$-classes or $H$-flux.[8]

In the same way that the metric can be decomposed into a metric on $N$ and a fibre contribution depending on $\xi^m$ as in (3.5), we can equally decompose the $H$ field into a base and fibre contribution,

$$H = \bar{H} + (\iota_m H) \wedge \xi^m + \frac{1}{2}(\iota_m \iota_n H) \wedge \xi^m \wedge \xi^n - \frac{1}{6}(\iota_m \iota_n \iota_p H) \wedge \xi^m \wedge \xi^n \wedge \xi^p, \quad (3.39)$$

where $\hat{\iota}_m \bar{H} = 0$ and $\widetilde{\iota}_m \bar{H} = 0$, so $\bar{H}$ is a basic tensor on $N$. Making use of (3.15), (3.24) and (3.28), this becomes

$$H = \bar{H} + \frac{1}{2\pi}\widetilde{F}_m \wedge \xi^m + \mathrm{d}B, \quad (3.40)$$

where

$$B = \frac{1}{2}B_{mn}\xi^m \wedge \xi^n. \quad (3.41)$$

The first Chern classes of the two $T^d$ bundles over $N$ described by connections $A^m$ and $\widetilde{A}_m$ are quantised, such that the integers

$$\frac{1}{2\pi}\int_{\Sigma^2} F^m \in \mathbb{Z}, \qquad \frac{1}{2\pi}\int_{\Sigma^2} \widetilde{F}_m \in \mathbb{Z} \quad (3.42)$$

describe the topology of the bundle $\hat{M}$. Since the NLSM on $\hat{M}$ is described by the metric $\hat{g}$ (whose only non-zero components are $g_{ij}$ and can be written as in (3.5)) and the closed 3-form $\hat{H}$ (whose only non-zero components $H_{ijk}$ are decomposed as (3.40)), the 1-forms $\xi^m$ and $\widetilde{\xi}_m$ fully characterise the topology of the NLSM on $\hat{M}$.

Note that the definition (3.34) ultimately depends on the 1-form connection $v_m$, related to $H$ locally by (3.15). Therefore, the doubled fibres on $\hat{M}$ encode not only the geometric data of $M$, but also information about the $b$-field configuration in the original NLSM. We will see that under the T-duality operation, the two sets of $T^d$ fibres will be exchanged which, intuitively speaking, swaps the geometric data and $b$-field data of the bundle $\hat{M}$.

## 3.3 T-duality in Non-Linear Sigma Models

As in the case of the compact boson, the T-dual model is derived by integrating out the (trivial) gauge fields $C^m$ first, and the fields $X^m$ corresponding to the original $T^d$ fibres second. This can be done because, even in the most general case, the action is still quadratic in $C^m$. To see this, let us examine the three terms in (3.30) separately:

---

[8]We note that both the 3-form $H$ and the 2-forms $\widetilde{F}_m$ are sometimes referred to as $H$-flux. The two quantities are, of course, related and the relevant object will be clear from context.

1. The metric term, using the expression (3.29) for the covariant derivatives, becomes

$$\hat{S}_{\text{metric}} = \frac{1}{2} \int_W g_{ij} dX^i \wedge \star dX^j - \int_W g_{ij} k_n^j dX^i \wedge \star C^n + \frac{1}{2} \int_W G_{mn} C^m \wedge \star C^n, \tag{3.43}$$

where $G_{mn}$ is defined in (3.6).

2. The $H$-flux term becomes

$$\hat{S}_{\text{flux}} = \frac{1}{3} \int_V H_{ijk} dX^i \wedge dX^j \wedge dX^k - \int_V H_{ijk} C^m k_m^i \wedge dX^j \wedge dX^k$$
$$+ \int_V H_{ijk} C^m k_m^i \wedge C^n k_n^j \wedge dX^k - \frac{1}{3} \int_V H_{ijk} C^m k_m^i \wedge C^n k_n^j \wedge C^p k_p^k. \tag{3.44}$$

Note that the Killing vectors are worldsheet scalars and we can commute them freely. Then, we can write this contribution in terms of the following contractions of $H$ with the Killing vectors,

$$\hat{S}_{\text{flux}} = \frac{1}{3} \int_V H_{ijk} dX^i \wedge dX^j \wedge dX^k - \int_V (\iota_m H)_{jk} C^m \wedge dX^j \wedge dX^k$$
$$+ \frac{1}{2} \int_V (\iota_n \iota_m H)_k C^m \wedge C^n \wedge dX^k - \frac{1}{6} \int_V (\iota_p \iota_n \iota_m H) C^m \wedge C^n \wedge C^p. \tag{3.45}$$

Substituting in (3.15), (3.24), and (3.28), we find

$$\hat{S}_{\text{flux}} = \frac{1}{3} \int_V H_{ijk} dX^i \wedge dX^j \wedge dX^k - \frac{1}{2\pi} \int_V (\mathrm{d}v_m)_{jk} C^m \wedge dX^j \wedge dX^k$$
$$+ \frac{1}{2} \int_V (\mathrm{d}B_{mn})_k C^m \wedge C^n \wedge dX^k. \tag{3.46}$$

3. Finally, in the Lagrange multiplier term one needs to be careful regarding the lift of the Killing vector to the doubled GNLSM. Equations (3.17), (3.21), and (3.29) imply

$$\hat{S}_{\text{lag}} = \frac{1}{2\pi} \int_V dC^m \wedge \left( v_{mi} dX^i - \iota_n v_m C^n - \Theta_{nm} C^n + d\hat{X}_m \right) \tag{3.47}$$

which, from (3.23), can be written

$$\hat{S}_{\text{lag}} = \frac{1}{2\pi} \int_V dC^m \wedge v_{mi} dX^i + \int_V B_{mn} dC^m \wedge C^n + \frac{1}{2\pi} \int_V dC^m \wedge d\hat{X}_m. \tag{3.48}$$

We now put together the three contributions. Taking care of the pull-back of exterior derivatives in target space to exterior derivatives in the worldsheet shows that the terms including $v$ and $B$ in $\hat{S}_{\text{flux}}$ and $\hat{S}_{\text{lag}}$ conspire to produce boundary

terms that can be written directly on $W = \partial V$. In conclusion, the action of the doubled GNLSM (3.30) is equivalent to

$$\hat{S} = S - \int_W C^m \wedge \star \left( g_{ij} k_m^j dX^i - \frac{1}{2\pi} v_{mi} \star dX^i - \frac{1}{2\pi} \star d\hat{X}_m \right)$$
$$+ \frac{1}{2} \int_W C^m \wedge \star \left( G_{mn} + B_{mn} \star \right) C^n , \tag{3.49}$$

where $S$ is the ungauged action (3.2). The linear coupling to the gauge field is through a current

$$\hat{J}_m = g_{ij} k_m^j dX^i - \frac{1}{2\pi} v_{mi} \star dX^i - \frac{1}{2\pi} \star d\hat{X}_m$$
$$= \hat{g}_{IJ} \hat{k}_m^J d\hat{X}^I - \frac{1}{2\pi} \hat{v}_{mI} \star d\hat{X}^I \tag{3.50}$$

It is apparent that the gauge fields $C^m$ appear only quadratically in the gauged action (3.49), and without derivatives, and so can be integrated out. This involves inverting the $(G_{mn} + B_{mn}\star)$ operator in the quadratic term, which is most easily done using a null coordinate system on the worldsheet $x^\pm$, where $\eta^{+-} = \epsilon^{+-} = 1$. In these coordinates, the action (3.49) becomes

$$\hat{S} = S - \int_W d^2\sigma \left( C_+^m \hat{J}_m^+ + C_-^m \hat{J}_m^- \right) + \int_W d^2\sigma \, C_+^m E_{mn} C_-^n , \tag{3.51}$$

where

$$E_{mn} = G_{mn} + B_{mn} . \tag{3.52}$$

We can now complete the square on the $C^m$ in (3.51) by a field redefinition

$$C^m = \widetilde{C}^m + C'^m , \tag{3.53}$$

where

$$\widetilde{C}_+^m = \hat{J}_n^- (E^{-1})^{nm}, \quad \widetilde{C}_-^m = (E^{-1})^{mn} \hat{J}_n^+ . \tag{3.54}$$

This is chosen such that, when substituted into (3.51), the terms which are linear in the $C^m$ all cancel, leaving

$$\hat{S} = S - \int_W d^2\sigma \, \hat{J}_m^- (E^{-1})^{mn} \hat{J}_n^+ + \int_W d^2\sigma \, C_+'^m E_{mn} C_-'^n . \tag{3.55}$$

Note that the $\widetilde{C}^m$ are determined fully in terms of the scalar fields $\hat{X}^I$ by (3.54), so that integrating out the $C^m$ is equivalent to integrating out the vectors $C'^m$. Furthermore, the $\widetilde{C}^m$ have the property that they transform in the same manner as the $C^m$, as in (3.12), under transformations (3.26) [46]. It follows that the $C'^m$ in (3.53) are globally-defined vectors on the worldsheet $W$. Since they appear only in the final term in (3.55), they are non-dynamical and can be integrated out to give

only an inconsequential prefactor in the path integral. Doing so leaves us with an action

$$\hat{S} = S - \int_W d^2\sigma \, \hat{J}_m^-(E^{-1})^{mn}\hat{J}_n^+ \,. \tag{3.56}$$

The geometry of the NLSM described by this action can be seen by writing the original action $S$ in (3.2) using null coordinates for $W$,

$$S = \int_W d^2\sigma \, \hat{\mathcal{E}}_{IJ}\partial_+\hat{X}^I\partial_-\hat{X}^J = \int_W d^2\sigma \, \mathcal{E}_{ij}\partial_+X^i\partial_-X^j \,, \tag{3.57}$$

where we have introduced the generalised metric

$$\hat{\mathcal{E}}_{IJ} = \hat{g}_{IJ} + \hat{b}_{IJ}, \quad \mathcal{E}_{ij} = g_{ij} + b_{ij} \,. \tag{3.58}$$

The equality in (3.57) is due to the lift of metric and $b$-field from $M$ to $\hat{M}$ in (3.18). It is important to note that $b$ is in general not a gauge-invariant quantity, and so neither is $\mathcal{E}$. When extracting the physical information of the sigma model, it will be important to go back to the metric $g$ and flux $H$.

Now, in the light-cone coordinates, the components of $\hat{J}_m$ are

$$\hat{J}_m^{\pm} = \left(\hat{g}_{IJ}\hat{k}_m^J \mp \frac{1}{2\pi}\hat{v}_{mI}\right)\partial_{\mp}\hat{X}^I \,, \tag{3.59}$$

so we can write the action $\hat{S}$ in (3.56) in a similar fashion to (3.57) as

$$\hat{S} = \int_W d^2\sigma \, \hat{\mathcal{E}}'_{IJ}\partial_+\hat{X}^I\partial_-\hat{X}^J \,, \tag{3.60}$$

where

$$\hat{\mathcal{E}}'_{IJ} = \hat{\mathcal{E}}_{IJ} - \left(\hat{g}_{IK}\hat{k}_m^K + \frac{1}{2\pi}\hat{v}_{mI}\right)(E^{-1})^{mn}\left(\hat{g}_{JL}\hat{k}_n^L - \frac{1}{2\pi}\hat{v}_{nJ}\right) \,. \tag{3.61}$$

When writing the original NLSM in the form (3.57), the metric and $b$-field determining its geometry can be recovered from the symmetric and antisymmetric parts of $\hat{\mathcal{E}}_{IJ}$ respectively. Similarly, the action (3.60) describes a NLSM on $\hat{M}$ whose metric and $b$-field are given by the symmetric and antisymmetric parts of $\hat{\mathcal{E}}'_{IJ}$. This can be neatly repackaged in terms of the globally-defined 1-forms $\xi^m$ and $\widetilde{\xi}_m$ introduced in (3.7) and (3.34) respectively,

$$\hat{\mathcal{E}}'_{IJ} = \hat{\mathcal{E}}_{IJ} - \left(\xi_I^p E_{pm} + \frac{1}{2\pi}\widetilde{\xi}_{mI}\right)(E^{-1})^{mn}\left(E_{nq}\xi_J^q - \frac{1}{2\pi}\widetilde{\xi}_{nJ}\right) \,. \tag{3.62}$$

Taking the symmetric and antisymmetric parts of this expression gives the metric and $b$-field of the dual NLSM as

$$\hat{g}' = g - G_{mn}\xi^m \otimes \xi^n + \frac{1}{(2\pi)^2}\widetilde{G}^{mn}\widetilde{\xi}_m \otimes \widetilde{\xi}_n \tag{3.63}$$

$$\hat{b}' = b - \frac{1}{2\pi}\widetilde{\xi}_m \wedge \xi^m - B + \frac{1}{(2\pi)^2}\widetilde{B} \,, \tag{3.64}$$

where

$$\widetilde{G}^{mn} = (E^{-1})^{(mn)}, \quad \widetilde{B}^{mn} = (E^{-1})^{[mn]} \tag{3.65}$$

and

$$\widetilde{B} = \frac{1}{2}\widetilde{B}^{mn}\widetilde{\xi}_m \wedge \widetilde{\xi}_n. \tag{3.66}$$

Now substituting in the form of the metric $g$ in (3.5), we find

$$\hat{g}' = \bar{g} + \frac{1}{(2\pi)^2}\widetilde{G}^{mn}\widetilde{\xi}_m \otimes \widetilde{\xi}_n, \tag{3.67}$$

Since the $b$-field is not globally-defined, it is better characterised by the flux $H$ which, from (3.40), is

$$\hat{H}' = \mathrm{d}b' = \bar{H} + \frac{1}{2\pi}F^m \wedge \widetilde{\xi}_m + \frac{1}{(2\pi)^2}\mathrm{d}\widetilde{B}. \tag{3.68}$$

Let us briefly comment on a specific term within the expression (3.64). From (3.31) and (3.37), the only term in $\hat{b}'$ which involves both the $X^m$ coordinates and the $\widetilde{X}_m$ coordinates lies in the $\widetilde{\xi}_m \wedge \xi^m$ term. In particular, there is a contribution to $\hat{b}'$ of the form

$$\frac{1}{2\pi}\,\mathrm{d}X^m \wedge \mathrm{d}\widetilde{X}_m \tag{3.69}$$

which contains both coordinates. Since this term is closed, it does not affect $\hat{H}'$ in (3.68). In any case, this term contributes trivially by $\exp(2\pi i\mathbb{Z})$ to the path integral on a worldsheet $W$ without boundary. This term will be crucial in the half-space gauging construction which we develop in the following section.

In summary, after integrating out the gauge fields $C^m$, we find an equivalent description of the original NLSM on $\hat{M}$ described by a target space with metric $\hat{g}'$ and closed 3-form $\hat{H}'$. We can then take the quotient by the action of the $\hat{k}_m$ (or, equivalently, gauge fix the $X^m$ = constant) to give a NLSM on a space which is a $T^d$ bundle over $N$ with metric $g' = \hat{g}'$ and closed 3-form $H' = \hat{H}'$, so the fibration is dictated by the connections $\widetilde{A}_m$ related to $\widetilde{\xi}_m$ by (3.37). Let us denote this space $M'$. Comparing (3.67) and (3.68) with (3.5) and (3.40), we see that the NLSM on $M'$ can be reached from the original NLSM on $M$ (3.2) by performing the following simple operations:

$$E \to \frac{1}{(2\pi)^2}E^{-1}, \quad \xi^m \leftrightarrow \widetilde{\xi}_m, \tag{3.70}$$

where the components of the matrix $E$ are defined in (3.52). Since both the original NLSM on $M$ and the dual description on $M'$ can be reached by different gauge choices in the parent theory on $\hat{M}$, they define the same quantum theory. We remark that due to the change of variables (3.36) the fields in the dual NLSM do not need to be directly equal to the Lagrange multipliers introduced in (3.30).

The operations (3.70) have a simple interpretation in the context of the doubled GNLSM on $\hat{M}$. We recall that $M$ is a $T^d$ fibration over $N$, described by connection

1-forms $A^m$ with curvature $F^m$ in (3.33). The larger space $\hat{M}$ is a $T^d$ fibration over $M$, described by connection 1-forms $\widetilde{A}_m$ with curvature $\widetilde{F}_m$ in (3.38). Since we also have $F^m = d\xi^m$ and $\widetilde{F}_m = d\widetilde{\xi}_m$, the operation (3.70) interchanges the Chern numbers $F^m$ with the $H$-classes $\widetilde{F}_m$, and so swaps the two sets of $T^d$ fibres.

## 3.4   Matching symmetries across T-dual NLSMs

A general NLSM (3.2) has two conserved currents. The first one is the Noether current associated to the isometry symmetry,

$$J_m = g_{ij}k_m^j dX^i - \frac{1}{2\pi}v_{mi} \star dX^i \,. \tag{3.71}$$

It is locally conserved thanks to the equations of motion. The second one is

$$\widetilde{J}^m = \frac{1}{2\pi} \star dX^m \,, \tag{3.72}$$

which is identically conserved off-shell, $d \star \widetilde{J}^m = 0$. In the case of the compact boson, they become the usual momentum and winding symmetries.

If the target space of the NLSM has non-contractible 1-cycles, it is well known what happens to these symmetries under T-duality: they are exchanged. If the target space does not have any non-contractible 1-cycles, then there are no winding modes, but the T-duality still exchanges the two currents $J$ and $\widetilde{J}$. This does not mean there are two conserved charges. The goal of this section is to understand the matching of symmetries in the modern language of topological operators, where the issue becomes completely transparent: when there are no non-trivial 1-cycles there is only one topological operator in each T-dual frame. In particular, if $H_1(M) \neq 0$, the operator defined by integrating the winding current is in fact not topological. In the T-dual model, it will translate to the putative topological operator for the isometry symmetry not being invariant under target space gauge transformations (and as a consequence also not topological in the worldsheet), and so not yielding a symmetry.[9]

To see this, consider the operator

$$U_m^{(w)}(\gamma_1) = \exp\left(iq_w \int_{\gamma_1} \star \widetilde{J}^m\right) = \exp\left(\frac{iq_w}{2\pi} \int_{\gamma_1} dX^m\right) \,, \tag{3.73}$$

where $\gamma_1$ is a 1-cycle in the worldsheet, and $q_w \in [0, 2\pi)$ is a parameter. Importantly, this operator is not topological since the integrand $dX^m$ is a closed but not exact 1-form, so integrating by parts using Stokes' theorem is not valid. This can be seen as follows.

For simplicity, let us consider the case of a single isometry $(d = 1)$. We have seen that whenever we have an isometry on $M$ acting without fixed points, we can

---

[9]For recent works exploring other subtleties in the definition of topological operators associated to Noether currents in Quantum Field Theory, see [65, 66].

describe the space as a fibration (3.4). This fibration may be trivial or non-trivial (i.e. the 1$^{\text{st}}$ Chern number associated with the connection $A$ in (3.31) can be zero or non-zero). In the former case, $M = N \times S^1$ is a product space and the $S^1$ is a non-trivial cycle, and we do have a winding charge and an associated topological operator (3.73). We are interested in the second case. Having a non-zero Chern number (which we take be 1 for simplicity here) implies that there exists no choice of globally-defined coordinates on the target space; if global coordinates did exist, we could use them to define a nowhere-vanishing global section, which would contradict having a non-zero 1$^{\text{st}}$ Chern number. As a consequence, there is a need for non-trivial transition functions between the coordinates $X_\alpha^m$ in each open patch $U_\alpha$, and the current $dX^m$ transforms not as a 1-form but as a connection.

Less abstractly, the non-topological nature of $U_m^{(w)}(\gamma_1)$ under deformations of $\gamma_1$ can be shown simply by finding configurations where $U_m^{(w)}(\gamma_1)$ depends explicitly on the choice of $\gamma_1$. This is straightforward: it suffices to take a particular configuration of the NLSM maps $\Phi(\sigma, \tau)$ where the string, say, begins wrapping the $S^1$ fibre at $\tau = 0$ and ends unwrapped at $\tau = 2\pi$. Then the charge measured by (3.73) is different for $\gamma_1 = (\sigma, \tau = 0)$ as opposed to $\gamma_1' = (\sigma, \tau = 2\pi)$, but these two choices of $\gamma_1$ are related by a smooth deformation and so $U_m^{(w)}(\gamma_1)$ cannot be topological. We will show a very explicit example of this, including how the global definition of the coordinates enters the game, for the case of $M = S^3$ in the following subsection.

In the T-dual model, the winding conserved current becomes the current associated with the dual isometry,[10]

$$J_m' = g_{ij}' \widetilde{k}_m^j d\widetilde{X}^i - \frac{1}{2\pi} \widetilde{v}_{mi} \star d\widetilde{X}^i \,, \tag{3.74}$$

where $\widetilde{v}_m$ are the components of a connection on $M'$ satisfying

$$\widetilde{\iota}_m H' = \frac{1}{2\pi} \mathrm{d}\widetilde{v}_m \,. \tag{3.75}$$

That is, the $\widetilde{v}_m$ take the role of the $v_m$ in the T-dual description of the NLSM. The exchange of Chern classes with $H$-classes (3.70) now implies that if in the initial model we had Chern number equal to 1, now we have an $H$-class of 1 in the T-dual frame. That is to say, $\widetilde{v}_m$ in (3.74) is also a connection with non-zero curvature and so transforms non-trivially under target space gauge transformations (in the same way as the $b$-field). In particular, when passing from one patch of the target space to another, one is forced to implement such a transformation. This spoils the topological property of the putative symmetry operator

$$\widetilde{U}_m^{(i)}(\gamma_1) = \exp\left(iq_i \int_{\gamma_1} \star J_m'\right), \tag{3.76}$$

---

[10]This can be seen from the equations of motion of the gauge fields $C^m$, and we will come back to it in section 4.6.

where $q_i \in [0, 2\pi)$. If we deform $\gamma_1$ in such a way that its image $\Phi(\gamma_1)$ moves between two patches in the target space $M$ then $\widetilde{v}_m$ can undergo a gauge transformation, discontinuously changing the value of the charge $\int_{\gamma_1} \star J'_m$ and, therefore, the operator $\widetilde{U}_m^{(i)}(\gamma_1)$. As a result, if $\widetilde{v}_m$ describes a non-trivial bundle the putative $U(1)$ isometry symmetry in this duality frame is broken. Again, we will show this explicitly in the next subsection in an example.

In the case where the Chern number or $H$-class is greater than 1, then there are discrete subgroups of the $U(1)$ isometry and winding symmetries which survive and are exchanged under duality.

## 3.5 Example: The 3-sphere as Hopf fibre vs $S^2 \times S^1$ with $H$-flux

In this subsection, we illustrate the general construction described throughout this section for the case of a NLSM with target space $M = S^3$ with no $H$-flux. As we will review, the T-dual model is a NLSM with target space $S^2 \times S^1$ with one unit of $H$-flux [45].

Consider one of the isometries of $S^3$. The metric can be written in the form (3.5) via the Hopf fibration $S^1 \hookrightarrow S^3 \twoheadrightarrow S^2$, giving the NLSM action

$$S = \frac{R^2}{8} \int_W (2d\phi - \cos\theta \, d\psi)^2 + d\theta^2 + \sin^2\theta \, d\psi^2 \,, \tag{3.77}$$

where the $S^3$ radius is $R$, and the fields are in the ranges $\theta \in (0, \pi)$, $\psi \in [0, 2\pi)$ and $\phi \in [0, 2\pi)$. Here, $\theta$ and $\psi$ are the coordinates on the $S^2$ base (i.e. the coordinates denoted by $Y^\mu$ in previous sections) while $\phi$ is the coordinate on the $S^1$ fibre (which was denoted by $X^m$ previously).[11] In the language of section 3, the Killing vector is

$$k = \frac{\partial}{\partial \phi} \,, \tag{3.78}$$

such that $G = g_{ij} k^i k^j = R^2$; that is, the radius of the $S^1$ fibre is also $R$. The 1-form $\xi$ dual to $k$ is

$$\xi = \mathrm{d}\phi - \frac{1}{2} \cos\theta \, \mathrm{d}\psi \,. \tag{3.79}$$

Writing this as in (3.31), we have

$$A = -\frac{1}{2} \cos\theta \, \mathrm{d}\psi \,, \tag{3.80}$$

which implies that $\frac{1}{2\pi} \int_{S^2} F = 1$, where $F = \mathrm{d}A$. That is, $S^3$ is described by a circle bundle over $S^2$ with first Chern number 1. This simple theory has $b = 0$, and so we can take $v = 0$ in (3.15).

---

[11]Note that the periodicities of these angles differ from standard texts and are chosen such that the fibre coordinate $\phi$ has period $2\pi$. Furthermore, we explicitly exclude the points $\theta = 0, \pi$ here, so the metric does not cover the full 3-sphere. This is to avoid subtle topological considerations which do not change the result of the computation. We will address these subtleties in detail below.

Let us now apply the T-duality methodology of section 3 to this example, with the aim of clarifying the technical aspects of the doubled GNLSM construction. That is, we wish to perform T-duality by viewing the NLSM (3.77) as a NLSM on the larger space $\hat{M}$ which is a $T^2$ fibre over $S^2$. The constraints (3.19) and (3.20) for the isometry on $\hat{M}$ to be gaugable are trivially satisfied in this example by choosing $\Theta = 0$. The doubled fibre direction in $\hat{M}$ will be denoted here by $\widetilde{\phi}$.[12]

Since the isometry is along the $\phi$ direction, the only derivative that gets promoted to a covariant derivative as in (3.29) is $d\phi \to D\phi = d\phi - C$. Furthermore, since $v = 0$, from (3.17) we have

$$\hat{v}_I D\hat{X}^I = d\widetilde{\phi}, \tag{3.81}$$

and so the doubled GNLSM action (3.30) becomes

$$\hat{S} = \frac{R^2}{8} \int_W \left[ d\theta^2 + d\psi^2 - 4\cos\theta \, d\psi \wedge \star D\phi + 4D\phi^2 \right] + \frac{1}{2\pi} \int_W C \wedge d\widetilde{\phi}. \tag{3.82}$$

Note that, as expected, it reduces to the usual gauging of the isometry that shifts $\phi$ by a constant together with a term involving the Lagrange multiplier $\widetilde{\phi}$ that forces the gauge field $C$ to be trivial.

Now we can proceed in an analogous way to the case of the compact boson described in section 2. We rewrite the action, which is quadratic in $C$, in the following fashion,

$$\hat{S} = \frac{R^2}{8} \int_W \left[ 4C^2 - 8C \wedge \star \left( d\phi - \frac{1}{2}\cos\theta \, d\psi - \frac{1}{2\pi R^2} \star d\widetilde{\phi} \right) \right.$$
$$\left. + d\theta^2 + d\psi^2 + 4d\phi^2 - 4\cos\theta \, d\psi \wedge \star d\phi \right]. \tag{3.83}$$

After completing the square and integrating out $C$ (which gives a normalization factor in the path integral), we get

$$\hat{S} = \frac{R^2}{8} \int_W \left( d\theta^2 + \sin^2\theta \, d\psi^2 \right) + \frac{1}{2(2\pi R)^2} \int_W d\widetilde{\phi}^2$$
$$- \frac{1}{4\pi} \int_W \cos\theta \, d\psi \wedge d\widetilde{\phi} + \frac{1}{2\pi} \int_W d\phi \wedge d\widetilde{\phi}. \tag{3.84}$$

In the first two terms we recognise the metric of $S^2 \times S^1$, where the $S^1$ now has radius $1/(2\pi R)$. The third term is a non-trivial $b$-field, namely

$$b' = -\frac{1}{4\pi} \cos\theta \, \mathrm{d}\psi \wedge \mathrm{d}\widetilde{\phi}, \tag{3.85}$$

which implies $\frac{1}{2\pi} \int_{S^2 \times S^1} H' = 1$, where $H' = \mathrm{d}b'$. The last term in (3.84) contributes a factor of $\exp(2\pi i \mathbb{Z})$ to the path integral, which we can safely disregard.

All in all, we conclude that the T-dual of the NLSM with target space $S^3$ and no $b$-field is a NLSM with target space $S^2 \times S^1$ and 1 unit of $H$-flux.

---

[12]This was denoted by $\hat{X}$ in previous sections. In this example, $\hat{X} = \widetilde{X}$ since $\Theta = 0$.

## Matching symmetries and global considerations

Let us illustrate the discussion of matching symmetries between T-dual frames in section 3.4 as explicitly as possible in this example. We will see the result described there: when the target space is a non-trivial $T^d$ bundle the winding current does not give rise to a topological operator, and in the dual frame the current associated with the isometry symmetry is also not topological due to target space gauge transformations. In order to see these subtle details, we must be careful about global properties of the $S^3$ geometry, which we discuss first.

The ranges of the various coordinates on the 3-sphere were specified below (3.77). However, there we chose the range of the coordinate $\theta \in (0, \pi)$ such that both North ($\theta = 0$) and South ($\theta = \pi$) poles of the 3-sphere are not covered by this chart. We did this because the coordinate $\psi$ is ill-defined in the poles, and while that does not cause issues in the part of the metric corresponding to the $S^2$ base (because $\sin \theta \to 0$), it is problematic in the Hopf connection.

If we want our sigma model to properly map onto the whole 3-sphere, we should consider two patches, which we denote $U_N$ and $U_S$. The Northern chart, $U_N$, excludes $\theta = \pi$ while the Southern chart, $U_S$, excludes $\theta = 0$. Then, the metric can be written

$$\mathrm{d}s^2 = \frac{R^2}{4} \left[ (2\mathrm{d}\phi - 2A)^2 + \mathrm{d}\theta^2 + \sin^2 \theta \mathrm{d}\psi^2 \right], \tag{3.86}$$

where $A$ is a basic connection with $\frac{1}{2\pi} \int_{S^2} F = 1$. We can take, for example, the potential

$$A_N = \frac{1}{2}(1 - \cos \theta)\mathrm{d}\psi, \quad A_S = \frac{1}{2}(-1 - \cos \theta)\mathrm{d}\psi \tag{3.87}$$

in the Northern and Southern charts respectively. The metric (3.86) is then well-defined provided that the coordinate $\phi$ has a non-trivial transition function such that its value in the Northern and Southern charts are related by

$$\phi_N = \phi_S - \psi \tag{3.88}$$

in the overlap $U_N \cap U_S$. The (globally-defined) 1-form $\xi$ dual to the Killing vector (3.78) is then

$$\xi = A + \mathrm{d}\phi, \tag{3.89}$$

with the connection $A$ given in each patch in (3.87).

This discussion emphasises the necessity of the non-trivial transition function (3.88). Let us now see how this implies that the winding current as defined in (3.72) does not give rise to a topological operator on the worldsheet. The would-be topological operator is given in (3.73) by (recall that we denote $X^m$ by $\phi$ in this example)

$$U^{(w)}(\gamma_1) = \exp\left(\frac{iq_w}{2\pi} \int_{\gamma_1} d\phi\right). \tag{3.90}$$

We will find a configuration of the NLSM map $\Phi(\sigma,\tau)$ on which $U^{(w)}(\gamma_1)$ explicitly depends on the choice of $\gamma_1$, and is therefore not topological. Consider a cylindrical worldsheet, described by coordinates $\tau \in [0,2\pi]$ and $\sigma \in [0,2\pi)$, with $\sigma \sim \sigma + 2\pi$. The relevant configurations are those in which the string begins wound around the $S^1$ fibre, and then unwinds from it. For example,

$$\phi_N(\sigma,\tau) = \sigma, \quad \theta(\sigma,\tau) = \frac{1}{2}\tau, \quad \psi(\sigma,\tau) = -\sigma. \tag{3.91}$$

At $\tau = 0$ the string begins at the North pole of the $S^2$ base and so must be described using the $\phi_N$ coordinate, which is wound around the $S^1$ fibre. At $\tau = 2\pi$, however, the string is at the South pole of the $S^2$ and so must be described with the $\phi_S$ coordinate which is, from (3.88), $\phi_S = 0$. That is, the string is unwound from the $S^1$ fibre as $\tau$ goes from 0 to $2\pi$.

Let us evaluate the operator $U^{(w)}(\gamma_1)$ in (3.90) when $\gamma_1$ wraps the periodic direction of the worldsheet at $\tau = 0$ and then at $\tau = 2\pi$. These two choices for $\gamma_1$ are related by a smooth deformation in the $\tau$ direction, so if $U^{(w)}(\gamma_1)$ were topological then it should give the same result for both. However, using the two different patches when computing (3.90) leads to different winding charges being measured:

$$U^{(w)}(\tau = 0) = \exp\left(\frac{iq_w}{2\pi}\int_{\tau=0} d\phi_N\right) = \exp\left(\frac{iq_w}{2\pi}\int_0^{2\pi} d\sigma\right) = \exp(iq_w), \tag{3.92}$$

$$U^{(w)}(\tau = 2\pi) = \exp\left(\frac{iq_w}{2\pi}\int_{\tau=2\pi} d\phi_S\right) = \exp\left(\frac{iq_w}{2\pi}\int_{\tau=2\pi} d\phi_N + d\psi\right) = 1. \tag{3.93}$$

In the dual model, the target space is $S^2 \times S^1$ with one unit of $H$-flux. The winding charge around the $S^1$ cycle matches the isometry charge of the original NLSM on $S^3$. On the other hand, the $b$-field profile specified by (3.85) leads to

$$\frac{1}{2\pi}d\widetilde{v} = \widetilde{\iota}H' = \frac{1}{4\pi}\sin\theta\,d\theta \wedge d\psi. \tag{3.94}$$

That is, $\widetilde{v}$ defines the components of a connection over $S^2$ with first Chern number 1.[13] Similarly to above, in the two patches of the $S^2$ we can choose

$$\widetilde{v}_N = \frac{1}{2}\left(1 - \cos\theta\right)d\psi, \quad \widetilde{v}_S = \frac{1}{2}\left(-1 - \cos\theta\right)d\psi. \tag{3.95}$$

The conserved charge, and the topological operator $\widetilde{U}_m^{(i)}(\gamma_1)$ (3.76) defined by integrating (3.74), explicitly depend on $\widetilde{v}$. Therefore, when $\gamma_1$ is deformed in such a way that $\Phi(\gamma_1)$ moves between the Northern and Southern patches of the target space, $\widetilde{U}_m^{(i)}(\gamma_1)$ changes discontinuously and so is not topological.

---

[13]In subsection 3.2, the $v_m$ defined a connection over $M$ which described the doubled space $\hat{M}$ as a $T^d$ bundle over $M$. From $v_m$, we defined $\widetilde{\xi}_m$ as in (3.34), such that $\widetilde{A}$ in (3.37) is a basic connection (i.e. it is defined on the base $N$). If there is only a single isometry ($d = 1$) and $\Theta = 0$, it follows from (3.34) and (3.37) that $v_m = \widetilde{A}_m$, since $\hat{X}_m = \widetilde{X}_m$. This is the case for the $S^3$ example studied here, and so $\widetilde{v}$ can be taken as a connection on the base $S^2$, rather than the full $S^2 \times S^1$.

# 4 Non-invertible defects in Non-Linear Sigma Models

We now have all the ingredients that we need in order to find non-invertible symmetries in more general NLSMs than the compact boson. The construction is similar to the one in section 2.3. We gauge a $U(1)$ isometry by coupling the NLSM to a worldsheet gauge field and constrain the gauge field to be flat with holonomies in $\mathbb{Z}_p$.[14] We use this $\mathbb{Z}_p$ gauging construction on one half of the worldsheet and take the original ungauged NLSM on the other. Then, for special values of the parameters, the gauged theory is equivalent to the original ungauged one, and the interface between the two halves of the worldsheet can be interpreted as a defect in a single theory (recall Figure 1).

For ease of presentation, we will first introduce the gauging construction in a worldsheet without boundary in subsection 4.1, and then proceed to discuss in detail the complications and modifications introduced by the presence of the boundary in the gauging procedure, adapting methods introduced in [61]. This is done in three steps: first we split the theory in two half-spaces in subsection 4.2. This requires the addition of a boundary term so that both halves are individually gauge invariant. Second, we rewrite the action of the NLSM with boundary in a manifestly gauge-invariant form and discuss its isometry symmetries in subsection 4.3. Third, we carry out the discrete gauging plus T-duality procedure in half-space in subsection 4.4. As we shall see, the only difference between our final result and the one of subsection 4.1 is the addition of a topological boundary term and this gives rise to the non-invertible fusion rules [40]. Equipped with this result, in subsection 4.5 we solve the self-duality conditions the NLSM has to satisfy in order to host the non-invertible defect. Finally, we discuss in detail the topological boundary term in subsection 4.6.

## 4.1 Discrete gauging in a full worldsheet

In this section, we discuss the gauging with gauge fields constrained to be flat with $\mathbb{Z}_p$ holonomy on the full worldsheet $W$ (without boundary). The formulae that arise are very similar to the T-duality construction but with extra factors of $p$, and reduce to the T-duality construction for $p = 1$. In the discussion of section 3.2, the following Lagrange multiplier term was added to the gauged sigma model action

$$\frac{1}{2\pi} \int_V dC^m \wedge d\hat{X}_m \,, \tag{4.1}$$

which is one of the terms in (3.48). Integrating out $\hat{X}_m$ imposes that the gauge field $C^m$ is flat with trivial holonomy. Here, we want a Lagrange multiplier term imposing that the gauge fields (which now we will denote by $c^m$) are $\mathbb{Z}_p$-valued. That is, they

---

[14]We recall that this can be viewed as gauging a $\mathbb{Z}_p$ subgroup of the U(1) isometry symmetry and then T-dualising, as in the case of the compact boson, discussed in section 2.2. We will sometimes refer to this construction as $\mathbb{Z}_p$ gauging.

must be flat, $dc^m = 0$, and have holonomies $\int_\sigma c^m \in \frac{2\pi}{p}\mathbb{Z}$ around 1-cycles $\sigma$ in the worldsheet. This is done by introducing a factor of $p$ in (4.1), which becomes

$$\frac{p}{2\pi} \int_V dc^m \wedge d\hat{X}_m \,. \tag{4.2}$$

Here, $\hat{X}_m$ remain periodic with period $2\pi$.[15] The geometry of the NLSM that emerges from integrating out $c^m$ is then similar to the T-dual geometry, except extra factors of $p$ will appear along the way.

The T-duality derivation discussed above starts with the doubled GNLSM (3.30). We now modify this by replacing $\hat{X}_m$ with $p\hat{X}_m$ to give the following action:

$$\hat{S}_{(p)} = \frac{1}{2} \int_W g_{ij} \, DX^i \wedge \star DX^j + \frac{1}{3} \int_V H_{ijk} \, DX^i \wedge DX^j \wedge DX^k$$
$$+ \frac{1}{2\pi} \int_V dc^m \wedge \hat{u}_{mI} D\hat{X}^I \,. \tag{4.3}$$

The only difference between this and (3.30) is that in the last term we have replaced $\hat{v}_m$ by a 1-form $\hat{u}_m$,

$$\hat{u}_m = v_m^\alpha + p \, d\hat{X}_m^\alpha \,, \tag{4.4}$$

where $\alpha$ labels an open patch $U_\alpha$ of the target space $M$. The condition that the 1-form $\hat{u}_m$ be globally-defined requires choosing the correct transition functions for $\hat{X}_m$, which therefore depend on $p$. As a result, changing $p$ changes the torus bundle over $M$. Contrasting this with $\hat{v}_m$ as defined in (3.17), we see that when expanding the final term of (4.3), the correct Lagrange multiplier term (4.2) appears.

The full action (4.3) can be written on $W$ by manipulations analogous to those leading to (3.49), giving

$$\hat{S}_{(p)} = S - \int_W c^m \wedge \star \left( g_{ij} k_m^j dX^i - \frac{1}{2\pi} v_{mi} \star dX^i - \frac{p}{2\pi} \star d\hat{X}_m \right)$$
$$+ \frac{1}{2} \int_W c^m \wedge \star \left( G_{mn} + B_{mn} \star \right) c^n \,. \tag{4.5}$$

As in the previous section, we now integrate out the gauge fields $c^m$ to yield a new NLSM. The factors of $p$ ensure that the result is *a priori* different from the original NLSM. The manipulations of section 3.3 to perform this integration go through without modification, so we will not repeat them. The result is the same as in (3.60)–(3.61) with the replacement $\hat{v} \to \hat{u}$. That is, the $\mathbb{Z}_p$ gauged theory has action

$$S'_{(p)} = \int_W d^2\sigma \, \hat{\mathcal{E}}_{IJ}^{(p)} \partial_+ \hat{X}^I \partial_- \hat{X}^J \,, \tag{4.6}$$

---

[15]The factor of $p$ could of course be absorbed into the definition of $\hat{X}_m$, which would then have period $2\pi/p$.

with

$$\hat{\mathcal{E}}_{IJ}^{(p)} = \hat{\mathcal{E}}_{IJ} - \left(\hat{g}_{IK}\hat{k}'^K_m + \frac{1}{2\pi}\hat{u}_{mI}\right)(E^{-1})^{mn}\left(\hat{g}_{JL}\hat{k}'^L_n - \frac{1}{2\pi}\hat{u}_{nJ}\right). \tag{4.7}$$

Here, $\hat{k}'_m$ are lifts of the original Killing vectors $k_m$ from $M$ to $\hat{M}$ given by

$$\hat{k}'_m = k_m + \frac{1}{p}\Theta_{mn}\frac{\partial}{\partial\hat{X}_n}. \tag{4.8}$$

These differ from $\hat{k}_m$ in (3.21) by an important factor of $1/p$ coming from the replacement $\hat{X}_m \to p\hat{X}_m$ which we are making to implement the $\mathbb{Z}_p$ gauging.

Recall that, in the previous section, the metric and $H$-flux of the dual NLSM were most easily identified when $\mathcal{E}'$ was written in terms of globally-defined quantities $\xi^m$ and $\widetilde{\xi}_m$. This was because these 1-forms decompose into a basic component (i.e. corresponding to a 1-form on $N$) and a fibre component, as in (3.31) and (3.37). In the present case of $\mathbb{Z}_p$ gauging, the $\xi^m$ remain globally defined and decomposable into base and fibre components, but the $\widetilde{\xi}_m$ need to be modified. The correct modification is simply to replace $\hat{v}_m$ in (3.34) by $\hat{u}_m$, so we define

$$\widetilde{\zeta}_m = \hat{u}_m + 2\pi B_{mn}\xi^n. \tag{4.9}$$

It follows from (3.23) that the $\widetilde{\zeta}_m$ are horizontal with respect to the $\hat{k}'_n$ (i.e. $\iota_{\hat{k}'_m}\widetilde{\zeta}_n = 0$). Therefore, they can be decomposed into a connection on the base of the target space, $N$, which we denote $\widetilde{a}_m$, and a component along the $\widetilde{X}_m$ fibres,

$$\widetilde{\zeta}_m = \widetilde{a}^\alpha_m + p\,\mathrm{d}\widetilde{X}^\alpha_m. \tag{4.10}$$

At a practical level, the decomposition (4.10) into base and fibre components will be useful when comparing the theories before and after the gauging. We denote the curvature of $\widetilde{a}_m$ by

$$\widetilde{f}_m = \mathrm{d}\widetilde{a}_m. \tag{4.11}$$

Inserting (3.7) and (4.9) into (4.7), we find that the NLSM after the $\mathbb{Z}_p$ gauging is described by (4.6) with

$$\hat{\mathcal{E}}_{IJ}^{(p)} = \hat{\mathcal{E}}_{IJ} - \left[E_{rm}\xi^r_I + \frac{1}{2\pi}\widetilde{\zeta}_{mI}\right](E^{-1})^{mn}\left[E_{ns}\xi^s_J - \frac{1}{2\pi}\widetilde{\zeta}_{nJ}\right]. \tag{4.12}$$

Expanding the generalised metric $\hat{\mathcal{E}}^{(p)}$ in (4.12) gives

$$\hat{\mathcal{E}}_{IJ}^{(p)} = \hat{\mathcal{E}}_{IJ} - E_{mn}\xi^m_I\xi^n_J + \frac{2}{2\pi}\xi^m_{[I}\widetilde{\zeta}_{J]m} + \frac{1}{(2\pi)^2}(E^{-1})^{mn}\widetilde{\zeta}_{mI}\widetilde{\zeta}_{nJ}. \tag{4.13}$$

As discussed in section 3.3, the symmetric and anti-symmetric parts of $\hat{\mathcal{E}}_{IJ}^{(p)}$ give the metric and $b$-field of the gauged NLSM on $\Gamma_+$:

$$\hat{g}^{(p)} = g - G_{mn}\xi^m \otimes \xi^n + \frac{1}{(2\pi)^2}\widetilde{G}^{mn}\widetilde{\zeta}_m \otimes \widetilde{\zeta}_n = \bar{g} + \frac{1}{(2\pi)^2}\widetilde{G}^{mn}\widetilde{\zeta}_m \otimes \widetilde{\zeta}_n, \tag{4.14}$$

$$\hat{b}^{(p)} = b + \frac{1}{2\pi}\xi^m \wedge \widetilde{\zeta}_m - B + \frac{1}{(2\pi)^2}\widetilde{\mathcal{B}}, \tag{4.15}$$

where

$$\widetilde{\mathcal{B}} = \frac{1}{2}\widetilde{B}^{mn}\widetilde{\zeta}_m \wedge \widetilde{\zeta}_n \tag{4.16}$$

and we have used (3.5) and (3.65). Taking the exterior derivative of $\hat{b}^{(p)}$ and using (3.40), we can write the gauge-invariant curvature $\hat{H}^{(p)}$ of the gauged NLSM on $\Gamma_+$ as

$$\hat{H}^{(p)} = \bar{H} + \frac{1}{2\pi}F^m \wedge \widetilde{\zeta}_m + \frac{1}{(2\pi)^2}\,\mathrm{d}\widetilde{\mathcal{B}}\,. \tag{4.17}$$

Let us briefly comment that, as for (3.69) in the T-duality derivation, there is a single term in $\hat{b}^{(p)}$ which involves both the $X^m$ and $\widetilde{X}_m$ coordinates. The full expression for $\hat{b}_{IJ}^{(p)}$ is given in (4.15) and contains a term proportional to $\widetilde{\zeta}_m \wedge \xi^m$. In particular, this term includes a contribution

$$\hat{b}^{(p)} = \frac{p}{2\pi}\,\mathrm{d}X^m \wedge \mathrm{d}\widetilde{X}_m + \cdots. \tag{4.18}$$

If the gauging is performed on the full worldsheet $W$, this term is inconsequential since it contributes trivially as $\exp(2\pi i p\mathbb{Z})$ to the path integral, but this will not be the case for worldsheets with boundary, as will be seen in the following subsections.

In general, (4.12) gives a NLSM that is different from the original one. We will be interested in the special cases in which (4.12) is the *same* NLSM as the original one. For the compact boson, this fixed the radius to be (2.12). For such self-dual models, the half-space construction with the original model in one half of the worldsheet and the dual one on the other will give the self-duality defect. The conditions for self-duality will be discussed in subsection 4.5.

## 4.2 Two half-spaces

The idea is to now split the worldsheet into two parts $\Gamma_-$ and $\Gamma_+$, separated by a curve $\gamma$ so that $W = \Gamma_- \cup \Gamma_+$ and $\partial\Gamma_+ = -\partial\Gamma_- = \gamma$. The NLSM is a theory of maps $\Phi : W \to M$ and $\gamma$ maps to a curve $\Phi(\gamma)$ in the target space. The map $\Phi$ can be viewed as a pair of maps $\Phi_\pm$ mapping each half $\Gamma_\pm$ into the target space $M$, with the condition that[16]

$$\Phi_+(\gamma) = \Phi_-(\gamma)\,. \tag{4.19}$$

The first step it to split the full action (3.1) on $W$ into two actions $S_\pm$ in $\Gamma_\pm$. This requires the addition of a boundary term involving a 1-form $a$ on $M$, so that

---

[16]Note that if the two maps $\Phi_\pm$ are smooth then the resulting map $\Phi : W \to M$ will not be smooth in general unless smoothness conditions are imposed at $\gamma$; in the path integral we consider integrations over all maps, not just smooth ones.

both are individually invariant under antisymmetric tensor gauge transformations

$$
\begin{aligned}
S &= \frac{1}{2} \int_W g_{ij} dX^i \wedge \star dX^j + \int_W \Phi^*(b) \\
&= \frac{1}{2} \int_{\Gamma_-} g_{ij} dX^i \wedge \star dX^j + \int_{\Gamma_-} \Phi_-^*(b) - \int_{\partial\Gamma_-} \Phi_-^*(a) \\
&\quad + \frac{1}{2} \int_{\Gamma_+} g_{ij} dX^i \wedge \star dX^j + \int_{\Gamma_+} \Phi_+^*(b) - \int_{\partial\Gamma_+} \Phi_-^*(a) \\
&= S_- + S_+ \,,
\end{aligned}
\tag{4.20}
$$

where we have used that $\partial\Gamma_+ = -\partial\Gamma_- = \gamma$ and defined

$$
S_\pm = \frac{1}{2} \int_{\Gamma_\pm} g_{ij} dX^i \wedge \star dX^j + \int_{\Gamma_\pm} \Phi_\pm^*(b) - \int_{\partial\Gamma_\pm} \Phi_\pm^*(a) \,.
\tag{4.21}
$$

Here, $a$ is a 1-form gauge field introduced such that (4.21) is invariant under

$$
b \to b + \mathrm{d}\lambda, \quad a \to a + \lambda \,.
\tag{4.22}
$$

Clearly, when the two terms $S_+$ and $S_-$ are added and the boundary condition (4.19) is imposed, the contributions from the 1-form gauge field $a$ cancel, leaving the original NLSM.

At the level of path integrals, the splitting of the NLSM into two pieces and, in particular, the splitting of the action (4.20), implies that there is a factorisation:

$$
\begin{aligned}
Z &= \int [d\Phi] e^{iS} \\
&= \int [d\Phi_-] e^{iS_-} \int_{\Phi_+(\gamma)=\Phi_-(\gamma)} [d\Phi_+] e^{iS_+} \\
&\equiv \int [d\Phi_-] e^{iS_-} \left( Z_+ \Big|_{\Phi_+(\gamma)=\Phi_-(\gamma)} \right),
\end{aligned}
\tag{4.23}
$$

where

$$
Z_+ = \int [d\Phi_+] e^{iS_+}
\tag{4.24}
$$

This can be re-expressed as follows. Let $\mathcal{C}$ be a curve in $M$ with the same topology as $\gamma$, so that if $\gamma$ is a circle, $\mathcal{C}$ is a closed curve. Then the path integral can be expressed as an integral over all maps $\Phi$ for which $\Phi(\gamma) = \mathcal{C}$, followed by an integral over curves $\mathcal{C}$ in $M$. The maps $\Phi$ for which $\Phi(\gamma) = \mathcal{C}$ can be split into maps $\Phi_\pm$ for which $\Phi_\pm(\gamma) = \mathcal{C}$. Then

$$
Z = \int [d\mathcal{C}] Z_+^{\mathcal{C}} Z_-^{\mathcal{C}}
\tag{4.25}
$$

where

$$
Z_\pm^{\mathcal{C}} = \int [d\Phi_\pm] e^{iS_\pm} \Big|_{\Phi_\pm(\gamma)=\mathcal{C}}
\tag{4.26}
$$

is the path integral over maps $\Phi_\pm : \Gamma_\pm \to M$ where $\Phi_\pm(\gamma) = \mathcal{C}$. From the QFT perspective, this factorisation of the path integral is a statement of locality.

## 4.3 Sigma Models with boundary

As we have just seen, in a NLSM on a worldsheet $\Gamma_+$ with boundary $\partial\Gamma_+$, the action (4.21) has a boundary term in order to have invariance under the $b$-field gauge transformations (4.22). Writing explicitly the pull-backs,

$$S_+ = \frac{1}{2}\int_{\Gamma_+} g_{ij}\,dX^i \wedge \star dX^j + \frac{1}{2}\int_{\Gamma_+} b_{ij}\,dX^i \wedge dX^j - \int_{\partial\Gamma_+} a_i dX^i. \qquad (4.27)$$

In the application of such a NLSM to String Theory, $g$ is a background metric, $b$ is a background antisymmetric gauge field and $a$ is a background abelian gauge field or "photon" corresponding to the endpoint of the string. For our purposes, it is important to keep track of the field $a$ and the coupled gauge transformations (4.22), as in general $b$ transforms by $b \to b + d\lambda$ under an isometry symmetry transformation.

Boundary conditions must be imposed on the NLSM fields $X^i$ at the boundary $\partial\Gamma_+ = \gamma$. Neumann conditions (corresponding to open strings) and Dirichlet conditions (corresponding to strings ending on a D-brane) are often considered. For Dirichlet boundary conditions, one needs to specify a submanifold $Y \subset M$ (the location of the D-brane) and the boundary is constrained to be mapped to $Y$, $\Phi_+(\gamma) \subseteq Y$. In that case, the NLSM should be viewed as a theory of maps $\Phi_+ : (\Gamma_+, \gamma) \to (M, Y)$. Here we will instead use the boundary conditions given by (4.19).

In sections 3 and 4.1 we have made extensive use of the fact that the WZ term of the NLSM (3.1) can be written as a 3-dimensional integral of (the pull-back of) $H$ on a 3-dimensional space $V$ whose boundary is the two-dimensional worldsheet, $\partial V = W$. This allowed us to deal with globally defined tensors, instead of directly with the $b$-field, which generally is a connection on a gerbe. This can be extended to the case with boundary as follows. We introduce a 2-surface $\Delta$ with boundary $\partial\Delta = -\partial\Gamma_+ = -\gamma$ so that $\Gamma_+ \cup \Delta$ is a closed 2-surface which is the boundary of a 3-space $V_+$, $\partial V_+ = \Gamma_+ \cup \Delta$, and extend $\Phi_+$ to a map $\Phi_+ : V_+ \to M$. For example, if $\Gamma_+$ is a hemisphere bounded by a circle $\partial\Gamma_+$, then $\Delta$ can be taken to be a disc with boundary the circle $-\partial\Gamma_+$, so that $V_+$ is a half-ball bounded by the hemisphere $\Gamma_+$ and disc $\Delta$ (see Figure 3).

The action (4.27) can then be rewritten as

$$S_+ = \frac{1}{2}\int_{\Gamma_+} g_{ij}\,dX^i \wedge \star dX^j + \frac{1}{3}\int_{V_+} H_{ijk}\,dX^i \wedge dX^j \wedge dX^k \wedge + \frac{1}{2}\int_{\Delta} \beta_{ij}\,dX^i \wedge dX^j$$

$$\qquad (4.28)$$

where

$$\beta = b - \mathrm{d}a. \qquad (4.29)$$

In the case of Dirichlet boundary conditions, the pull-back $i^*H$ (where $i$ is the inclusion map of $Y$ in $M$, $i : Y \to M$) is required to be exact, so that $i^*H = \mathrm{d}b\big|_Y$ for some globally-defined $b\big|_Y$ on $Y$, so that $\beta$ is a well-defined 2-form on $\Phi_+(\Delta)$

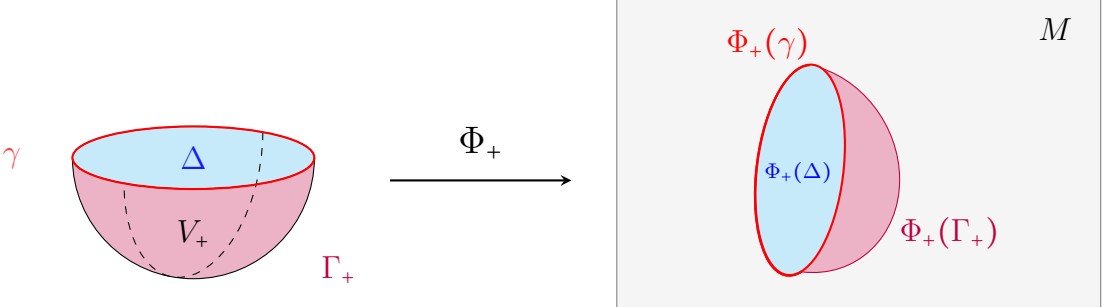

**Figure 3**. Geometry of a NLSM on a worldsheet $\Gamma_+$ with boundary, $\Phi_+ : (\Gamma_+, \gamma) \to (M, \Phi_+(\gamma) = \Phi_-(\gamma))$. There exists a disc $\Delta$ with image $\Phi_+(\Delta)$, and a 3-volume $V_+$, that satisfy $\partial V_+ = \Gamma_+ \cup \Delta$ such that $\partial \Gamma_+ = -\partial \Delta = \gamma$.

[61]. For a given NLSM, this constrains the possible submanifolds $Y$ at which the D-brane can be located. For Dirichlet boundary conditions, $a$ and $\beta$ can be taken to be defined on $Y$ rather than on the whole of $M$. The action then gives a well-defined contribution to the path integral (i.e. independent of the auxiliary choices of $V_+$ and $\Delta$) if $\frac{1}{2\pi}H$ represents an integral cohomology class of $M$ (as before) and also $\frac{1}{2\pi}\beta$ represents an integral cohomology class of $Y$. This can be expressed in terms of relative cohomology [61].

In general, $\beta$ is a locally defined 2-form on $M$ and is a gerbe connection rather than a 2-form. This implies that the integral $\int_\Delta \Phi_+^*(\beta)$ needs to be defined carefully, using the methods of [63]. However, for contractible $\Delta$, as in the example of a disc considered above, $b\big|_{\Phi_+(\Delta)}$ and hence $\beta\big|_{\Phi_+(\Delta)}$ can be taken to be well-defined 2-forms on $\Phi_+(\Delta)$, so that $\int_\Delta \beta_{ij} dX^i \wedge dX^j$ is well-defined in the usual way.

Let us now move on to studying the symmetries of the Sigma Model with boundary. Suppose the target space $M$ has a $U(1)^d$ isometry generated by Killing vectors (3.8) satisfying (3.10). Then the action given by (4.27) or (4.28) is invariant under (3.9) up to a boundary term,

$$\delta S_+ = \int_{\partial \Gamma_+} \alpha^m \left( \frac{1}{2\pi} v_m + \iota_m \beta \right). \tag{4.30}$$

There will be full invariance, including the boundary term, if this vanishes. A sufficient condition for this is

$$\frac{1}{2\pi} v_m + \iota_m \beta = \frac{1}{2\pi} dh_m, \tag{4.31}$$

for some functions $h_m$ on $M$. We note that there is a freedom to shift the $h_m$ by constants. Also, (4.31) implies that $\beta$ is an invariant 2-form on $M$,

$$\mathcal{L}_m \beta = 0. \tag{4.32}$$

For Dirichlet boundary conditions, it is sufficient that this is satisfied on $Y$. Then, with $\beta$ a 2-form on $Y$, the condition is that

$$\frac{1}{2\pi}v_m\big|_Y + \iota_m\beta = \frac{1}{2\pi}\mathrm{d}h_m\,, \tag{4.33}$$

for some functions $h_m$ on $Y$, which again implies (4.32). The gauging of the isometry symmetry for the NLSM with boundary and with Dirichlet boundary conditions was given in [61] and is reviewed in appendix B.

## 4.4 Gauging and dualising in half-space

We now wish to consider the theory obtained by replacing $Z_+^{\mathcal{C}}$ in (4.25) by $\hat{Z}_{+(p)}^{\mathcal{C}}$, which is obtained from the original theory on $\Gamma_+$ by gauging a $(\mathbb{Z}_p)^d$ subgroup of the $U(1)^d$ isometry group and then T-dualising.[17] This is achieved by coupling to 1-form gauge fields $c^m$ and then adding a Lagrange multiplier term

$$\frac{p}{2\pi}\int_{\Gamma_+} c^m \wedge d\hat{X}_m\,. \tag{4.34}$$

Up to boundary terms, the construction is the same as that in section 4.1. For Dirichlet boundary conditions, the gauging with boundary was given by [61] and is reviewed in appendix B. The net result is the addition of a boundary term depending on the gauge field

$$\frac{1}{2\pi}\int_\gamma \Phi_+^*(h_m)c^m \tag{4.35}$$

where $h_m$ is the function on $Y$ arising in (4.31).

In our case the boundary conditions are instead $\Phi_+(\gamma) = \mathcal{C}$. For any given $\mathcal{C}$ it could be viewed as a special case of this Dirichlet construction (with $Y = \mathcal{C}$). However, we want to integrate over all possible curves $\mathcal{C}$, so the constraints on $Y = \mathcal{C}$ need to be replaced by conditions required on the whole of $M$. For rigid isometry symmetry, we require (3.10) and (4.31) on all of $M$ instead of the condition (4.33) on $Y$, and $h_m$ is a function on $M$. The gauging is similar to that in section 4.1, but one further condition is needed for the theory with boundary:

$$\mathcal{L}_m h_n = 0\,. \tag{4.36}$$

Given these conditions, the correct coupling of the gauge fields $c^m$ can be written

$$\begin{aligned}
\hat{S}_{+(p)} = S_+ &- \int_{\Gamma_+} c^m \wedge \star \left(g_{ij}k_m^j dX^i - \frac{1}{2\pi}v_{mi}\star dX^i - \frac{p}{2\pi}\star d\hat{X}_m\right) \\
&+ \frac{1}{2}\int_{\Gamma_+} c^m \wedge \star\,(G_{mn} + B_{mn}\star)\,c^n + \frac{1}{2\pi}\int_\gamma \Phi_+^*(h_m)c^m\,.
\end{aligned} \tag{4.37}$$

---

[17]T-duality for open strings and sigma-models with boundary has been discussed previously in [67–69] for Dirichlet and Neumann boundary conditions.

At this stage, recall that in order to define a topological defect (or, more generally, a topological interface) we should impose the topological boundary condition (1.1) which imposes that the gauge fields $c^m$ vanish on $\gamma$. Therefore the last term in (4.37) vanishes, and we are left with

$$
\begin{aligned}
\hat{S}_{+(p)} = S_+ &- \int_{\Gamma_+} c^m \wedge \star \left( g_{ij} k_m^j dX^i - \frac{1}{2\pi} v_{mi} \star dX^i - \frac{p}{2\pi} \star d\hat{X}_m \right) \\
&+ \frac{1}{2} \int_{\Gamma_+} c^m \wedge \star \left( G_{mn} + B_{mn} \star \right) c^n \, ,
\end{aligned}
\tag{4.38}
$$

which is of precisely the same form as (4.5) with the only differences being that 1) the couplings to the gauge fields $c^m$ are only on $\Gamma_+$, and 2) the original action $S_+$ contains a boundary term as in (4.21). Note that (4.36) is not needed for gauge invariance with these boundary conditions. Note also that the same result would have been obtained with weaker boundary conditions in which only the component of $c$ that is tangential to $\gamma$ is required to vanish on $\gamma$, i.e. the pull-back of $c$ to the boundary is zero. In this case, the gauge parameters $\alpha^m$ satisfy the Dirichlet boundary conditions that they are constant on $\gamma$.

We can now integrate out the gauge fields $c^m$ in precisely the same manner as in section 4.1. The result is that the gauged NLSM on $\Gamma_+$ is described by the following action

$$
S'_{+(p)} = \int_{\Gamma_+} d^2\sigma \, \hat{\mathcal{E}}^{(p)}_{IJ} \partial_+ \hat{X}^I \partial_- \hat{X}^J - \int_\gamma \Phi^*_+(a) \, ,
\tag{4.39}
$$

with $\hat{\mathcal{E}}^{(p)}$ given by (4.12).

## 4.5 Conditions for duality defects

We are finally in a position to identify duality defects. Let us take stock. We have split the worldsheet $W$ into two sections, $\Gamma_-$ and $\Gamma_+$. On $\Gamma_-$, we have a NLSM with action $S_-$ defined in (4.21). This contains the terms of the original NLSM (3.2) as well as a 1-form gauge field $a$ on the boundary $\gamma = -\partial\Gamma_-$. On $\Gamma_+$, we have a similar setup but we have gauged a $\mathbb{Z}_p$ subgroup of the U(1) isometry symmetry, resulting in a theory described by (4.39). We note that when we add $S_- + S'_{+(p)}$, the boundary terms involving the 1-form gauge field $a$ cancel, leaving a NLSM on $\Gamma_-$ described by the generalised metric $\hat{\mathcal{E}}_{IJ}$ (as in (3.57)) and a NLSM on $\Gamma_+$ described by the generalised metric $\hat{\mathcal{E}}^{(p)}_{IJ}$ in (4.12),

$$
S'_{(p)} = \int_{\Gamma_-} d^2\sigma \, \mathcal{E}_{ij} \partial_+ X^i \partial_- X^j + \int_{\Gamma_+} d^2\sigma \, \hat{\mathcal{E}}^{(p)}_{IJ} \partial_+ \hat{X}^I \partial_- \hat{X}^J \, .
\tag{4.40}
$$

A non-invertible defect will be present if we are able to fix parameters such that the theories in the bulk of $\Gamma_-$ and $\Gamma_+$ describe the same NLSM. For such a choice of parameters, the boundary $\gamma$ between the two halves of the worldsheet can then be thought of as a topological operator in a single bulk theory.

What are the conditions for duality defects? Naïvely, it would seem that we should demand $\hat{\mathcal{E}}_{IJ} = \hat{\mathcal{E}}_{IJ}^{(p)}$, but this is too restrictive. There are two subtleties to account for. First, recall that the generalised metric $\hat{\mathcal{E}}_{IJ}$ contains both the metric and $b$-field (as in (3.58) for the original ungauged NLSM), so the metric and $b$-field of the gauged NLSM on $\Gamma_+$ are given by the symmetric and anti-symmetric parts of $\hat{\mathcal{E}}_{IJ}^{(p)}$ respectively. Now, while $\hat{g}_{IJ}^{(p)} \equiv \hat{\mathcal{E}}_{(IJ)}^{(p)}$ is a well-defined tensor, $\hat{b}_{IJ}^{(p)} \equiv \hat{\mathcal{E}}_{[IJ]}^{(p)}$ is not (rather, it is a gerbe connection). The well-defined tensor which describes the $b$-field is its flux $\hat{H}^{(p)} = \mathrm{d}\hat{b}^{(p)}$. Therefore, it is sufficient to impose that the metric $\hat{g}^{(p)}$ and flux $\hat{H}^{(p)}$ match between the two sides of the worldsheet.

Secondly, we recall from section 3 that, after gauging the isometry symmetry, the resulting NLSM is naturally formulated using the $(Y^\mu, \widetilde{X}_m)$ coordinates.[18] The original NLSM was formulated using the $(Y^\mu, X^m)$ coordinates. In the case of T-duality in section 3.3, this arose because all dependence of the generalised metric of the gauged theory, $\hat{\mathcal{E}}'_{IJ}$, on the original fibre coordinates $X^m$ disappeared, with the exception of a topological term (3.69) that did not contribute to the path integral. The same happens in the present context, when gauging a $\mathbb{Z}_p$ subgroup on $\Gamma_+$, and the same topological term appears in $\hat{b}^{(p)}$ (with a factor of $p$, see (4.18)). Since $\Gamma_+$ has a boundary, its contribution cannot be neglected: it can be written as a boundary term that gives a TQFT living on the defect

$$S_\gamma = \frac{p}{2\pi} \int_\gamma X^m d\widetilde{X}_m \,. \tag{4.41}$$

Because this lives only on $\gamma$, it does not preclude the theory in the *bulk* of $\Gamma_-$ and the *bulk* of $\Gamma_+$ from being the same after gauging in $\Gamma_+$. This generalises the TQFT (2.15) found on the non-invertible defect in the compact boson CFT to general NLSMs. Aside from this boundary term, there is no remaining dependence on the $X^m$ coordinates in $\hat{\mathcal{E}}_{IJ}^{(p)}$ (see (4.14) and (4.17)). To ensure that the two halves describe the same NLSM, the conditions to impose are

$$\hat{g}^{(p)} = \hat{g}\Big|_{X^m \to \widetilde{X}_m}, \quad \hat{H}^{(p)} = \hat{H}\Big|_{X^m \to \widetilde{X}_m}, \tag{4.42}$$

which say that the metric and $H$-flux are the same on the two sides of the defect, up to a renaming of coordinates $X^m \leftrightarrow \widetilde{X}_m$. The metric and curvature $H$ of the original NLSM on $\Gamma_-$ are given in (3.5) and (3.40), while the metric and curvature of the $\mathbb{Z}_p$ gauged NLSM on $\Gamma_+$ are given in (4.14) and (4.17). Inserting these expressions into the constraints (4.42) immediately gives

$$\widetilde{a}_m = pA^m, \quad G_{mn} = \left(\frac{p}{2\pi}\right)^2 \widetilde{G}^{mn}, \quad \widetilde{F}_m = pF^m, \quad B_{mn} = \left(\frac{p}{2\pi}\right)^2 \widetilde{B}^{mn}, \tag{4.43}$$

where the condition $\widetilde{a}_m = pA^m$ is understood to hold up to gauge transformations. These constraints can be more succinctly understood as follows:

---

[18]Due to the change of coordinates (3.36), recall that the $\widetilde{X}_m$ are not in general equal to the Lagrange multipliers $\hat{X}_m$.

- The constraints involving the various connections can be phrased in terms of their field strengths,

$$pF^m = \widetilde{F}_m = \widetilde{f}_m. \tag{4.44}$$

This is a constraint on the topology. The Chern class and $H$-class of the original NLSM are exchanged and scaled by $p$ during the gauging procedure, and this constraint ensures that the NLSM after the gauging has the same topological data.

- The other constraints can be written in terms of the torus moduli $E_{mn}$ (defined in (3.52)),

$$E_{mn} = \left(\frac{p}{2\pi}\right)^2 (E^{-1})^{mn}. \tag{4.45}$$

This is a metric constraint. It generalises the self-dual radius of the compact boson (2.12) and we see that, for general NLSMs with WZ term, the notion of the radius of the $S^1$ is replaced by the length of the orbit of the Killing vectors and the $b$-field flux along the fibres.

For the simplest case $d = 1$, where only a single isometry of the target space is gauged, we necessarily have $B_{mn} = 0$ and the matrix $G_{mn}$ becomes a scalar $G = g_{ij}k^i k^j$ which is just the norm squared of the Killing vector corresponding to the isometry under consideration. In this case, the conditions (4.44) and (4.45) become

$$pF = \widetilde{F} = \widetilde{f} \tag{4.46}$$

and

$$G = \frac{p}{2\pi}. \tag{4.47}$$

The latter is precisely the same as the condition (2.12) on the radius of the compact boson (where $G = R^2$).

At this point, it is straight-forward to generalise this result to the gauging of a subgroup

$$\prod_{m=1}^{d} \mathbb{Z}_{p_{(m)}} \subseteq U(1)^d \tag{4.48}$$

of the isometry symmetries. The self-duality conditions in this case are

$$p_{(m)} F^m = \widetilde{F}_m = \widetilde{f}_m, \qquad E_{mn} = \frac{p_{(m)}}{2\pi} \frac{p_{(n)}}{2\pi} (E^{-1})^{mn}, \tag{4.49}$$

with no sum over $m$ or $n$ in any of the expressions. These expressions reduce to (4.44) and (4.45) when $p_{(m)} = p$ for all $m$. These conditions are fairly restrictive, but there are many NLSMs where our non-invertible defects are possible; we explore this in some examples in section 5.

## 4.6 TQFT on the boundary and equations of motion

In this subsection, we study the role of the TQFT (4.41) which lives on the defect after we impose the self-duality conditions described in the previous subsection. In particular, we will see that this TQFT yields the correct equations of motion on both sides of the defect, similarly to the discussion in section 2.3, generalising [40].

Before studying the effect of the TQFT, let us return to the theory where a $\mathbb{Z}_p$ subgroup of the isometry symmetry has been gauged on half-space $\Gamma_+$. The relations between the isometry and winding currents on either side of the defect can be derived by studying the equations of motion of the gauge fields $c^m$. Before integrating them out, the gauged NLSM can be written as in (4.38). Since now we are only concerned with local relations between the fields in a neighbourhood of the defect, we can make use of the freedom to shift $v_m$ by an exact 1-form (as mentioned below equation (3.23)) to set $\Theta_{mn} = 0$ locally. Then we can set $\eta_m = 0$ in (3.36) and find that the two sets of coordinates can be identified, $\hat{X}_m = \widetilde{X}_m$.

The equations of motion for the gauge fields $c^m$ are then

$$(G_{mn} + B_{mn}\star)c^n = g_{ij}k_m^j dX^i - \frac{1}{2\pi}v_{mi} \star dX^i - \frac{p}{2\pi} \star d\widetilde{X}_m \qquad (4.50)$$

in $\Gamma_+$. Recall that, in order for the half-space gauging to define a topological defect, we imposed Dirichlet boundary conditions on the gauge fields: $c^m\big|_\gamma = 0$. Therefore, (4.50) implies that on the defect we have[19]

$$\left(g_{ij}k_m^j dX^i - \frac{1}{2\pi}v_{mi} \star dX^i\right)\bigg|_\gamma = \frac{p}{2\pi} \star d\widetilde{X}_m\bigg|_\gamma. \qquad (4.51)$$

We now discuss how these same relations between the dual symmetries arise by studying the theory in the presence of the non-invertible defect carrying the TQFT (4.41). The action of the theory with the defect inserted along the 1-cycle $\gamma$ in the worldsheet can be written as

$$S'_{(p)} = S_+ + S_\gamma + S_-, \qquad (4.52)$$

where $S_+$ and $S_-$ are given in (4.21), and $S_\gamma$ is given in (4.41). Since we impose the self-duality conditions, the metric $g$ and curvature $H$ are the same on both sides of the worldsheet.

There are several independent contributions to the equations of motion in the presence of the defect. Firstly, there are contributions from the bulk of the left- and

---

[19]This has interpretation of matching the Noether current (3.71) associated with the isometry symmetry of the original NLSM on one side of the defect to the dual isometry current (3.72) of the gauged theory on the other side, where the fields are the $(Y^\mu, \widetilde{X}_m)$. Indeed, this generalises the situation for T-duality outlined in subsection 3.4 (which is recovered when $p = 1$). As highlighted there, we emphasise that the existence of locally conserved currents does not necessarily lead to well-defined topological operators due to topological obstructions.

right-hand sides of the worldsheet. These give the same bulk equations of motion as the original NLSM on the worldsheet without boundary. The more interesting contributions come from terms on $\gamma$. These terms arise from the variation of $S_\gamma$ and also from boundary terms in the variations of $S_+$ and $S_-$. In particular, the variation of the action under $X^m \to X^m + \delta X^m$ has contributions on $\gamma$ of the form

$$\delta S'_{(p)}\Big|_\gamma = \int_\gamma \left( -g_{ij}k_m^j \star dX^i + \frac{1}{2\pi}v_{mi}dX^i + \frac{p}{2\pi}d\widetilde{X}_m \right) \delta X^m. \qquad (4.53)$$

It follows that the equation of motion for $X^m$ on $\gamma$ is precisely the same as (4.51), giving an equality between the Noether current for the isometry symmetry on one side of the defect and the current for the winding symmetry on the other. Studying the variation of the action under $\widetilde{X}_m \to \widetilde{X}_m + \delta\widetilde{X}_m$ gives an analogous equality between the winding current on one side and the Noether current associated with the isometry on the other.

One interesting remark is that the topological term (4.41) on $\gamma$ is always the same, regardless of the details of the sigma model. As explained in [40], the Tambara-Yamagami fusion rules can be recovered from this defect Lagrangian, and so we see that they will be the same in all NLSMs admitting such a defect. This was indeed the expected result, as there are general arguments why any self-duality defect should satisfy this fusion [14, 15].

## 5 Examples

Let us spell out the construction above explicitly in some examples. We begin in subsection 5.1 with the case of a 3-sphere, which is one of the simplest non-trivial cases in the sense that, while there is no winding so as to directly import the construction of [37], we are still able to construct a non-invertible defect. In subsection 5.2 we show that this extends to all odd dimensional spheres. Other sigma models that admit our non-invertible defects are Lens spaces, nilfolds and Wess-Zumino-Witten models, which we treat in subsections 5.3, 5.4 and 5.5.

### 5.1 The 3-sphere with $H$-flux

We once again describe the 3-sphere as the Hopf fibration, namely as an $S^1$ fibre over $S^2$ with first Chern number equal to 1. As we have seen in section 4, Chern numbers and $H$-classes are exchanged (up to factors of $p$) when gauging $\mathbb{Z}_p$ subgroups of isometries. Therefore, in order to find a NLSM which is invariant under such a gauging, we need to start with a NLSM with non-zero $H$-flux. Therefore, let us begin with a NLSM with action

$$S_{\Gamma_-} = \frac{R^2}{8} \int_{\Gamma_-} \left[ (2d\phi - \cos\theta\, d\psi)^2 + d\theta^2 + \sin^2\theta\, d\psi^2 \right] - \frac{\kappa}{4\pi} \int_{\Gamma_-} \cos\theta\, d\psi \wedge d\phi, \qquad (5.1)$$

on the first half of the worldsheet $\Gamma_-$. Here, $\kappa \in \mathbb{Z}$ measures the flux of the $b$-field and is quantised such that the WZ term is well-defined.

Having analysed the general case in the previous section, we could use the results (4.44)–(4.45) directly. Instead, we will perform the path integral gauging manipulations explicitly with the hope that this will clarify any technical aspects of the general construction. We know from subsection 4.4 that all subtleties regarding the presence of a boundary (including the boundary contribution to the gauge variation of (5.1)) do not affect the Lagrangian of the $\mathbb{Z}_p$ gauged theory, which is identical to the Lagrangian describing the gauging of the $\mathbb{Z}_p$ subgroup of the isometry symmetry on a worldsheet without boundary (thanks to the topological boundary condition $c|_\gamma = 0$). Therefore, we will proceed through this example as if $\Gamma_\pm$ were worldsheets without boundaries.

The conventions for the $S^3$ coordinates $(\theta, \psi, \phi)$ are the same as used in section 3.5. As in that section, the Killing vector is given by (3.78) and the 1-form dual to it is $\xi$ in (3.79). The closed 3-form $H = \mathrm{d}b$ is

$$H = \frac{\kappa}{4\pi} \sin\theta \, \mathrm{d}\theta \wedge \mathrm{d}\psi \wedge \mathrm{d}\phi \,, \tag{5.2}$$

which is proportional to the $S^3$ volume form and satisfies

$$\iota H = \frac{\kappa}{4\pi} \sin\theta \, \mathrm{d}\theta \wedge \mathrm{d}\psi = \frac{1}{2\pi} \mathrm{d}v \,, \tag{5.3}$$

so a choice of $v$ is

$$v = -\frac{\kappa}{2} \cos\theta \, \mathrm{d}\psi \,, \tag{5.4}$$

and, therefore, the NLSM (5.1) has $\kappa$ units of $H$-flux,

$$\frac{1}{2\pi} \int_{S^2} \widetilde{F} = \kappa \,. \tag{5.5}$$

We now wish to gauge a $\mathbb{Z}_p$ subgroup of the global symmetry associated with the isometry (3.78) on the other half of the worldsheet, $\Gamma_+$. We denote the gauge field by $c$. The $\mathbb{Z}_p$ gauged NLSM on $\hat{M}$ is given by (4.38) in general. In this case, it reduces to a similar expression to (3.82), with an extra factor of $p$ in the Lagrange multiplier term,

$$\hat{S}_{+(p)} = \frac{R^2}{8} \int_{\Gamma_+} \left[ d\theta^2 + d\psi^2 - 4\cos\theta \, d\psi \wedge \star(d\phi - c) + 4(d\phi - c)^2 \right]$$
$$- \frac{\kappa}{4\pi} \int_{\Gamma_+} \cos\theta d\psi \wedge (d\phi - c) + \frac{p}{2\pi} \int_{\Gamma_+} c \wedge d\widetilde{\phi} \,. \tag{5.6}$$

The action is still quadratic in $c$, and can be rewritten as

$$\hat{S}_{+(p)} = \frac{R^2}{2} \int_{\Gamma_+} c^2 - 2c \wedge \star \left( d\phi - \frac{1}{2}\cos\theta \, d\psi + \frac{\kappa}{4\pi R^2}\cos\theta \star d\psi - \frac{p}{2\pi R^2} \star d\widetilde{\phi} \right)$$
$$+ \frac{R^2}{8} \int_{\Gamma_+} \left( 4d\phi^2 + d\psi^2 + d\theta^2 - 4\cos\theta \, d\phi \wedge \star d\psi \right) - \frac{\kappa}{4\pi} \int_{\Gamma_+} \cos\theta \, d\psi \wedge d\phi \,, \tag{5.7}$$

Many simplifications take place after integrating out $c$. The final result is

$$S'_{+(p)} = \int_{\Gamma_+} \left[ \frac{p^2}{32\pi^2 R^2} \left( 2d\widetilde{\phi} - \frac{\kappa}{p} \cos\theta d\psi \right)^2 + \frac{R^2}{8} \left( d\theta^2 + \sin^2\theta \, d\psi^2 \right) \right]$$
$$- \frac{p}{4\pi} \int_{\Gamma_+} \cos\theta \, d\psi \wedge d\widetilde{\phi} + \frac{p}{2\pi} \int_{\Gamma_+} d\phi \wedge d\widetilde{\phi}. \tag{5.8}$$

It is now easy to see which self-duality conditions need to be imposed such that the resulting NLSM on $\Gamma_+$ describes an $S^3$ target space with the same $H$-flux as the original NLSM on $\Gamma_-$. Comparing the coefficients in the Hopf connection and $b$-field terms in (5.1) and (5.8) leads to

$$p = \kappa. \tag{5.9}$$

This is the topological constraint (4.44). Lastly, matching the radius of the $S^1$ fibre we find

$$R^2 = \frac{p}{2\pi}. \tag{5.10}$$

This is the constraint on the norm of the Killing vectors (4.47). In subsection 5.5, we will see that this condition is equivalent to the quantization condition which makes the NLSM (5.1) conformal. That is, when (5.10) is satisfied, the NLSM (5.1) is equivalent to the $SU(2)_\kappa$ WZW model.

As in the case of the compact boson, the last term in (5.8), which gives a trivial contribution to the path integral when the worldsheet has no boundary, gives now a contribution that becomes the non-trivial TQFT on the defect,

$$S_\gamma = \frac{p}{2\pi} \int_\gamma \phi \, d\widetilde{\phi}. \tag{5.11}$$

## 5.2   Odd spheres

The above construction can be straightforwardly generalised to higher dimensional spheres. For even dimensional $S^{2n}$, it is generally the case that isometries act with fixed points, and so one of the technical assumptions of our construction does not hold. For $S^{2n+1}$, however, there exists a fibration

$$S^1 \hookrightarrow S^{2n+1} \twoheadrightarrow \mathbb{CP}^n, \tag{5.12}$$

with first Chern number equal to 1. Suppose we also turn on $\kappa$ units of $H$-flux through a non-trivial 2-cycle in the $\mathbb{CP}^n$ (which exists since $H_2(\mathbb{CP}^n) \neq 0$). Then the self-duality conditions under a $\mathbb{Z}_p$ gauging impose that

$$\kappa = p, \quad R^2 = \frac{p}{2\pi}, \tag{5.13}$$

where $R$ is the radius of the sphere.

## 5.3 Lens spaces

Another generalisation of the example in subsection 5.1 are Lens spaces, namely quotients $S^3/\mathbb{Z}_q$. These can also be written as fibrations

$$S^1 \hookrightarrow S^3/\mathbb{Z}_q \twoheadrightarrow S^2 \,, \tag{5.14}$$

with first Chern number equal to $q$. Including as well a Wess-Zumino term giving $\kappa$ units of $H$-flux, the self-duality conditions become

$$pq = \kappa, \quad R^2 = \frac{p}{2\pi} \,. \tag{5.15}$$

## 5.4 Tori and nilfolds

The simplest geometries which are usually discussed in the context of T-duality are tori. Let us see how our $\mathbb{Z}_p$ gauging construction can be applied in the example of a 3-torus of radius $R$. Let us denote the three $2\pi$-periodic coordinates on the $T^3$ target space by $X, Y, Z$. We consider a single isometry generated by $k = \partial_Z$. The torus is a product space and, therefore, can be expressed as a trivial $S_Z^1$ bundle over $T_{XY}^2$ with first Chern number 0 (the subscripts denote the coordinates on each submanifold).[20] A non-trivial $b$-field can be turned on whose curvature is proportional to the $T^3$ volume form:

$$H = \frac{\kappa}{(2\pi)^2} \, \mathrm{d}X \wedge \mathrm{d}Y \wedge \mathrm{d}Z \,. \tag{5.16}$$

We then find

$$\frac{1}{2\pi} \, \mathrm{d}v = \iota H = \frac{\kappa}{(2\pi)^2} \, \mathrm{d}X \wedge \mathrm{d}Y \,, \tag{5.17}$$

so the $H$-flux is

$$\frac{1}{2\pi} \int_{T_{XY}^2} \widetilde{F} = \kappa \,. \tag{5.18}$$

Clearly, the condition (4.44) in this case imposes $\kappa = 0$, and the self-duality construction is not possible for non-zero $\kappa$. If $\kappa = 0$, so there is no $b$-field, then (4.44) is satisfied for all $p$ and (4.47) simply imposes (2.12). Of course, the NLSM on $T^3$ without $H$-flux is equivalent to three copies of the compact boson studied in section 2 and this defect is the same as the one studied there for the compact scalar $Z$.

A more interesting situation arises if, instead of a simple torus, we consider a non-trivial $S^1$ bundle over $T^2$. For example, let us consider a target space $M$ given by the principal fibre bundle

$$S_Z^1 \hookrightarrow M \twoheadrightarrow T_{XY}^2 \,, \tag{5.19}$$

with first Chern number $n > 0$. This is an example of a nilmanifold. We again consider the isometry generated by the Killing vector $k = \partial_Z$ around the $S^1$ fibre,

---

[20]In the language of the previous sections, the base coordinates are $Y^\mu = (X, Y)$ and the fibre coordinate is $X^m = Z$.

which gives

$$\frac{1}{2\pi} \int_{T_{XY}^2} F = n \,. \tag{5.20}$$

We can turn on the same $b$-field as before, with curvature (5.16), such that there are again $\kappa$ units of $H$-flux as in (5.18). The self-duality condition (4.44) then imposes

$$pn = \kappa \,, \tag{5.21}$$

while the constraint (4.47) again fixes the radius of the $S^1$ fibre as in (2.12). That is, a NLSM on the nilmanifold with first Chern number $n$ and $pn$ units of $H$-flux is self-dual under gauging a $\mathbb{Z}_p$ subgroup of the isometry symmetry around the $S^1$ fibre, and so hosts a non-invertible duality defect.

## 5.5 WZW models

We now consider a large family of NLSMs where the target space is a compact simply-connected semi-simple Lie group $G$, with Lie algebra $\mathfrak{g}$. For $G = \mathrm{SU}(N)$, we will find that imposing the self-duality constraints (4.44)–(4.45) under $\mathbb{Z}_p$ gauging of a single isometry fixes the model to be the $\mathrm{SU}(N)_p$ Wess-Zumino-Witten (WZW) model.

In this subsection we will denote the maps defining the NLSM by $g : W \to G$ (instead of $\Phi$) to conform with standard convention. The manifold can be parametrised by taking $g$ in a unitary representation, $r$, of $G$, whose generators we denote $t_A^{(r)}$ with $A = 1, \ldots, \dim G$. Our conventions for generators will match those of [70], i.e. the Lie algebra generators will be Hermitian and satisfy

$$[t_A, t_B] = i f_{AB}{}^C t_C \,. \tag{5.22}$$

Then the non-degenerate Killing form

$$K(X, Y) = \frac{1}{2h^\vee} \mathrm{Tr}(\mathrm{ad}(X)\mathrm{ad}(Y)) \,, \tag{5.23}$$

has components

$$K(t_A, t_B) = -\frac{1}{2h^\vee} f_{AC}{}^D f_{BD}{}^C = \delta_{AB} \,, \tag{5.24}$$

where $h^\vee$ is the dual Coxeter number of $\mathfrak{g}$. This can be used to raise and lower Lie algebra indices. In a representation $r$, the normalisation of the trace is then fixed to

$$\mathrm{Tr}_r(t_A^{(r)} t_B^{(r)}) = 2x_r \delta_{AB} \,, \tag{5.25}$$

where $t_A^{(r)}$ are the generators of the representation and $x_r$ is the Dynkin index of the representation.

The left- and right-invariant Maurer-Cartan 1-forms are given by

$$L = g^{-1}\,\mathrm{d}g = i L_i^A\,\mathrm{d}X^i\,t_A^{(r)} \,, \qquad R = \mathrm{d}g\,g^{-1} = i R_i^A\,\mathrm{d}X^i\,t_A^{(r)} \,, \tag{5.26}$$

respectively. The Maurer-Cartan equations $\mathrm{d}L + \frac{1}{2}[L, L] = 0$ and $\mathrm{d}R - \frac{1}{2}[R, R] = 0$ then imply that the 1-forms $L^A = L_i^A \, \mathrm{d}X^i$ and $R^A = R_i^A \, \mathrm{d}X^i$ satisfy

$$\mathrm{d}L^A - \frac{1}{2}f^A{}_{BC}\, L^B \wedge L^C = 0, \qquad \mathrm{d}R^A + \frac{1}{2}f^A{}_{BC}\, R^B \wedge R^C = 0 \,. \tag{5.27}$$

There is then a left- and right-invariant metric with components

$$g_{ij} = \frac{1}{\lambda^2} L_i^A L_j^B \delta_{AB} = \frac{1}{\lambda^2} R_i^A R_j^B \delta_{AB} \,, \tag{5.28}$$

where $\lambda^2$ is a positive coupling constant. The components $L_i^A$ and $R_i^A$ can be understood as two different vielbeins on the group manifold. The 3-form $H$ is taken to be

$$H = \frac{\kappa}{24\pi} f_{ABC} L^A \wedge L^B \wedge L^C = \frac{\kappa}{24\pi} f_{ABC} R^A \wedge R^B \wedge R^C \,, \tag{5.29}$$

where $\kappa \in \mathbb{Z}$, such that $\frac{1}{2\pi}[H]$ defines an integral cohomology class [62].

The action of the NLSM is taken to be (3.2) with $g$ and $H$ given by (5.28) and (5.29) respectively. This action can also be written

$$S = -\frac{1}{4\lambda^2 x_r} \int_W \mathrm{Tr}_r \left( g^{-1} \, \mathrm{d}g \wedge \star g^{-1} \, \mathrm{d}g \right) + \frac{\kappa}{24\pi x_r} \int_V \mathrm{Tr}_r \left( g^{-1} \, \mathrm{d}g \wedge g^{-1} \, \mathrm{d}g \wedge g^{-1} \, \mathrm{d}g \right) \,. \tag{5.30}$$

The 1-forms $L^A$ and $R^A$ are dual to left- and right-invariant Killing vector fields

$$k_A^L = L_A^i \frac{\partial}{\partial X^i}, \qquad k_A^R = R_A^i \frac{\partial}{\partial X^i} \,, \tag{5.31}$$

whose Lie brackets read

$$\left[ k_A^L, k_B^L \right] = i f_{AB}{}^C k_C^L, \quad \left[ k_A^R, k_B^R \right] = i f_{AB}{}^C k_C^R, \quad \left[ k_A^L, k_B^R \right] = 0 \,. \tag{5.32}$$

The isometry group of the Lie group manifold is, therefore, $G \times G$. At the level of the group element $g \in G$, the two factors of $G$ in the isometry group simply act by left- and right-multiplication respectively.

Let us now apply the results of previous sections to this case. Firstly, the full isometry group of the WZW is, in general, non-abelian and in order to perform the gauging operations that we have discussed in previous sections we must select a non-anomalous abelian subgroup of the isometries. The largest abelian subgroup is generated by Killing vectors $\{k_r^L, k_r^R\}$ where $r = 1, \ldots, \mathrm{rank}(\mathfrak{g})$ which span the Cartan subalgebras of the left and right isometry groups. These generate a $\mathrm{U}(1)^{2\,\mathrm{rank}(\mathfrak{g})}$ subgroup of isometries. For simplicity, we focus on gauging a single $\mathrm{U}(1)$ isometry and take $G = \mathrm{SU}(N)$ for the remainder of this section. Without loss of generality, we select a particular Killing vector

$$k = \frac{\sqrt{2}}{\lambda^2} k_\star^L \equiv \frac{\partial}{\partial \phi} \,, \tag{5.33}$$

where $\star$ denotes some particular value from 1 to rank$(\mathfrak{g})$. This normalisation of $k$ ensures that $\phi$ is $2\pi$-periodic.[21] In general, the normalisation depends on the choice of representation $r$ which $g$ transforms in, as well as the group $G$. The normalisation given here is relevant for $G = \mathrm{SU}(N)$ and $g$ in the fundamental representation, for which $x_{\mathrm{fund}} = 1/2$.

Using the techniques developed in the previous section, we now analyse the gauging of a $\mathbb{Z}_p$ subgroup of this $\mathrm{U}(1)$ isometry on half of the worldsheet, and impose the self-duality conditions discussed in section 4.5. For the Killing vector (5.33), we find

$$G = g_{ij}k^i k^j = \frac{2}{\lambda^2}\,, \tag{5.34}$$

$$\xi = \frac{1}{\sqrt{2}}L_\star\,, \tag{5.35}$$

$$\iota_k H = \frac{\kappa\sqrt{2}}{4\pi}\,\mathrm{d}L_\star\,. \tag{5.36}$$

Comparing (5.36) with (3.15), we have

$$v = \frac{\kappa}{\sqrt{2}}L_\star + \chi \tag{5.37}$$

for any closed 1-form $\chi$. In general, to couple background fields to this $\mathrm{U}(1)$ we are allowed to introduce a $\Theta$ parameter as in (3.21) and the $\mathrm{U}(1)$ isometry symmetry will be non-anomalous if (3.20) is satisfied. In this example it is possible to find a solution for $v$ where $\Theta = 0$. In this case, the condition for the 't Hooft anomaly to vanish, (3.20), reduces to $\iota_k v = 0$. In other words, in order for the symmetry to be non-anomalous, we must find a closed 1-form $\chi$ such that $v$ in (5.37) satisfies $\iota_k v = 0$. This can be done as follows. From (5.35) and (3.7), $L_{\star i}$ has components

$$L_{\star i} = \sqrt{2}G^{-1}g_{ij}k^j = \frac{\lambda^2}{\sqrt{2}}g_{i\phi}\,. \tag{5.38}$$

Let us denote coordinates on $G$ by $(Y^\mu, \phi)$ where, as described in section 3.1, $G$ can be seen as a $\mathrm{U}(1)$ bundle over a base $N$ on which the $Y^\mu$ are coordinates. In these coordinates,

$$L_\star = \frac{\lambda^2}{\sqrt{2}}g_{\mu\phi}\,\mathrm{d}Y^\mu + \sqrt{2}\,\mathrm{d}\phi\,. \tag{5.39}$$

It follows that choosing $\chi = -\kappa\,\mathrm{d}\phi$ results in

$$v = \frac{\kappa}{\sqrt{2}}\left(L_\star - \sqrt{2}\,\mathrm{d}\phi\right), \tag{5.40}$$

which satisfies $\iota_k v = 0$ and so the isometry is non-anomalous.

---

[21]This coordinate was denoted $X^m$ in previous sections. Here, we use $\phi$ for later comparison with the results of section 5.1.

The NLSM with target space $G$ is then specified by the two classes $[F]$ and $[\widetilde{F}]$, with

$$F = \mathrm{d}\xi = \frac{1}{\sqrt{2}} \mathrm{d}L_\star \,, \quad \widetilde{F} = \mathrm{d}v = \frac{\kappa}{\sqrt{2}} \mathrm{d}L_\star \,. \tag{5.41}$$

Having calculated all the relevant quantities, we are in a position to apply the self-duality constraints discussed in section 4.5. The condition (4.44) sets

$$p = \kappa \,, \tag{5.42}$$

while (4.45) imposes that

$$\lambda^2 = \frac{4\pi}{p} = \frac{4\pi}{\kappa} \,. \tag{5.43}$$

As is well-known, this is also the relation between the coupling $\lambda^2$ and $\kappa$ for which this NLSM is conformal [62]. For this choice of $\lambda^2$, the NLSM (5.30) is the $\mathrm{SU}(N)$ WZW model at level $\kappa$. This result implies that a $\mathrm{SU}(N)_\kappa$ WZW model is self-dual under gauging a $\mathbb{Z}_\kappa$ subgroup of the $\mathrm{U}(1)$ global symmetry associated to any one of its isometries, and there is a corresponding non-invertible symmetry.

## $\mathrm{SU}(2)_\kappa$ WZW model

Throughout this work, and particularly in section 5.1, we have studied the NLSM with target space $S^3$ as an example. Since $S^3$ is the group manifold of $\mathrm{SU}(2)$, given the result of the previous subsection we now see that when the duality conditions are satisfied, the NLSM with target space $S^3$ is an alternative parameterisation of the WZW model with target space $\mathrm{SU}(2)$ at level $\kappa$. The two descriptions can be related by parametrising the map $g : W \to \mathrm{SU}(2)$ by

$$g(\theta, \psi, \phi) = e^{-i\psi t_3/\sqrt{2}} e^{-i\theta t_1/\sqrt{2}} e^{+i\sqrt{2}\phi t_3} \,, \tag{5.44}$$

where $t_A = \frac{1}{\sqrt{2}}\sigma_A$ and $\sigma_A$ are the Pauli matrices. The choice of normalisation is such that these fundamental $\mathrm{SU}(2)$ generators are indeed normalised as in (5.25). The structure constants are then $f_{ABC} = \sqrt{2}\epsilon_{ABC}$. The angles take values in $\theta \in (0, \pi)$, $\psi, \phi \in [0, 2\pi)$. In this parameterisation, the metric (5.28) agrees with (3.77) provided that we associate

$$\lambda^2 = \frac{4}{R^2} \,. \tag{5.45}$$

Furthermore, the WZ term (5.29) precisely matches (5.2). This demonstrates explicitly that the NLSM (5.1) with parameters related as in (5.9) and (5.10) is an equivalent parameterisation of the $\mathrm{SU}(2)_\kappa$ WZW model.

## Verlinde lines

The WZW models, for compact semi-simple $G$, define rational CFTs (RCFTs) and so come with an associated set of topological lines known as the Verlinde lines [71], which we will denote by $\mathcal{L}_j$. These lines are in one-to-one correspondence with chiral

primary operators, which are themselves labelled by integrable unitary representations of the chiral algebra. The fusion rules of these lines are the same as that of the chiral primaries, and is given explicitly by the Verlinde formula [71]. It is natural to ask whether the duality defects which we have constructed above in $\mathrm{SU}(2)_\kappa$ WZW models could be a subset of the Verlinde lines. The simplest check is to analyse the fusion of the Verlinde lines at level $\kappa$ and try to determine a subset of the lines with Tambara-Yamagami fusion (2.13).

For $\mathfrak{su}(2)_\kappa$, the Verlinde lines are labelled by $j = 0, \frac{1}{2}, 1, \ldots, \frac{\kappa}{2}$ and the fusion rules are [72]

$$\mathcal{L}_{j_1} \otimes \mathcal{L}_{j_2} = \bigoplus_{\substack{j=|j_1-j_2|, \\ j+j_1+j_2 \in \mathbb{Z}}}^{j_{\max}} \mathcal{L}_j \,, \tag{5.46}$$

where $j_{\max} = \min(j_1 + j_2, \kappa - j_1 - j_2)$. At level $\kappa = 1$, there are are two Verlinde lines: $\mathcal{L}_0$ and $\mathcal{L}_{1/2}$. Their fusion, according to (5.46), is

$$\mathcal{L}_0 \otimes \mathcal{L}_0 = \mathcal{L}_0, \quad \mathcal{L}_0 \otimes \mathcal{L}_{1/2} = \mathcal{L}_{1/2} \otimes \mathcal{L}_0 = \mathcal{L}_{1/2}, \quad \mathcal{L}_{1/2} \otimes \mathcal{L}_{1/2} = \mathcal{L}_0 \,, \tag{5.47}$$

so $\mathcal{L}_0 = 1$ is a trivial line and $\mathcal{L}_{1/2}$ is a $\mathbb{Z}_2$ generator. If we identify $\eta = \mathcal{L}_0$ and $\mathcal{N} = \mathcal{L}_{1/2}$, these fusion rules are the same as those of (2.13) for $p = 1$. This is, of course, a trivial case where all the lines are invertible but nevertheless the Tambara-Yamagami fusion of the duality defects aligns with the Verlinde line fusion rules.

At level $\kappa = 2$, there are three Verlinde lines: $\mathcal{L}_0$, $\mathcal{L}_{1/2}$, and $\mathcal{L}_1$. Again, the line $\mathcal{L}_0 = 1$ is trivial, while the other lines have fusion rules given by

$$\mathcal{L}_{1/2} \otimes \mathcal{L}_{1/2} = \mathcal{L}_0 \oplus \mathcal{L}_1, \quad \mathcal{L}_{1/2} \otimes \mathcal{L}_1 = \mathcal{L}_1 \otimes \mathcal{L}_{1/2} = \mathcal{L}_{1/2}, \quad \mathcal{L}_1 \otimes \mathcal{L}_1 = \mathcal{L}_0 \,. \tag{5.48}$$

In this case, associating $\eta = \mathcal{L}_1$ and $\mathcal{N} = \mathcal{L}_{1/2}$, we see that the Verlinde line fusion is precisely the Tambara-Yamagami fusion of the duality defects shown in (2.13).

This pattern stops at level $\kappa = 3$. At this level, the Verlinde line fusion rules (5.46) do not contain any line which generates a $\mathbb{Z}_3$ subalgebra, and so cannot contain a subset of lines satisfying the $TY(\mathbb{Z}_3)$ fusion rules in (2.13) with $p = 3$. In general, the duality defects for $\mathrm{SU}(N)_\kappa$ are not Verlinde lines. By definition, this means that they do not commute with the full chiral algebra and so will, in general, act differently on different states within a chiral module.

## 6 Summary and outlook

In this work, we have constructed non-invertible symmetries in a large class of Non-Linear Sigma Models. This was done by gauging a discrete subgroup of an abelian isometry symmetry, T-dualising and then finding the self-dual values of the various parameters. To understand how general these defects are, let us review the technical assumptions which were necessary to perform the half-space gauging, as well as the conditions that self-duality imposes on the NLSM.

The *a priori* conditions on the theory are very mild:

- We need the worldsheet $W$ to be orientable, so that we can define the WZ term, and without boundary (or with a boundary with suitable boundary conditions), so that the only contribution from the topological term arising in the gauging procedure is at the location of the defect. We have chosen to present our analysis in Lorentzian signature and the Lorentzian formulae presented throughout are best adapted to the case of a cylindrical worldsheet. This can be analytically continued to a Euclidean cylinder and our discussion also applies to more general worldsheets – we could have worked in Euclidean signature from the start and then the gauging can be explicitly performed on any Riemann surface $W$. We require, however, that there is a curve $\gamma$ (possibly disconnected) which divides $W$ into two parts.

- The target space needs to have an isometry that acts without fixed points. Otherwise, the matrix $G_{mn}$ will have zero eigenvalues at the fixed points and will not be invertible there; this interferes with the derivation of T-duality as well as the more general discrete gauging of the isometry.[22]

- These isometries need to give rise to global symmetries of the NLSM with WZ term both for a worldsheet with or without boundary; in particular, the flux $H = \mathrm{d}b$ and the 2-form $\beta$ introduced in section 4.4 need to have vanishing Lie derivatives along the Killing vectors. Moreover, the groups that we want to gauge need to be free of 't Hooft anomalies, which leads to various constraints discussed in sections 3.2 and 4.4.

If these conditions are met, we can gauge a discrete subgroup of the isometries in half of the worldsheet $\Gamma_+ \subset W$, and we have checked in section 4.4 that the subtleties introduced by the presence of a boundary can be dealt with. In general, this will produce a different NLSM on $\Gamma_+$. The main result of our work is the identification of the self-duality conditions that must be satisfied by the NLSM in order for the gauged theory on $\Gamma_+$ to be the same as the original theory. If it is, the theory admits a non-invertible symmetry. We reproduce here the result (4.49) for convenience of the reader,

$$ p_{(m)} F^m = \widetilde{F}_m = \widetilde{f}_m, \qquad E_{mn} = \frac{p_{(m)}}{2\pi} \frac{p_{(n)}}{2\pi} (E^{-1})^{mn}, \tag{6.1} $$

where all the relevant quantities are defined in sections 3.1, 3.2 and 4.1; and the $p_{(m)}$ are the orders of the various factors in the $\prod_m \mathbb{Z}_{p_{(m)}} \subset \mathrm{U}(1)^d$ subgroup that we are gauging. The two conditions in (6.1) are imposing 1) that the topological data of

---

[22]There are some cases where a T-duality can be derived even if the isometries have fixed points, see e.g. [73, 74].

the NLSM is unchanged by the gauging, and 2) that we are sitting at the self-dual value of the torus moduli.

Interestingly, we have found that a TQFT living on the non-invertible defect arises explicitly by keeping track of boundary terms in the half-space gauging procedure. This TQFT is always a BF-type theory and is responsible for the non-invertible $\text{TY}(\mathbb{Z}_p)$ fusion rules (2.13) of the defect [40].

Note that the topology of the worldsheet plays an important role: if $W$ has no non-trivial 1-cycles, any flat gauge field configuration will automatically be trivial. Then there is no gauge field holonomy to constrain and the gauging construction reduces to a T-duality, so that the theory on $\Gamma_+$ is the T-dual of the one on $\Gamma_-$ and the result is a T-fold [46, 64, 75]. The self-duality condition then restricts the NLSM to one that is self-dual under T-duality (for the compact boson, this sets the circle to have the self-dual radius) and the theory on $\Gamma_+$ is the same as the one on $\Gamma_-$, with no defect at $\gamma$. Indeed, that is also expected from the fusion rules of the TY category (2.13), where the defect $\mathcal{N}$ turns out to be invertible if the topology of the worldsheet is trivial.

While the conditions (6.1) are fairly restrictive, we have found that there are many examples of NLSMs which satisfy them, including spheres, Lens spaces and nilfolds, for particular values of the radius and $b$-field flux. A particularly interesting class of examples where we can explicitly construct self-duality defects are WZW models. We find the surprising result that, starting with a NLSM whose target space is a Lie group $G$, the required conditions for self-duality coincide precisely with the quantization of the coupling that makes the NLSM exactly conformal [62] (see (5.43)). The precise relation of the level $\kappa$ with the discrete subgroup which must be gauged to give a self-duality of the theory depends on various group-theoretic quantities: our computation in subsection 5.5 shows that $\text{SU}(N)_\kappa$ is self-dual under under a $\mathbb{Z}_\kappa$ gauging and so hosts a topological defect line with $\text{TY}(\mathbb{Z}_\kappa)$ fusion, without any restrictions on the level and rank. It is straightforward to see that the same normalisations will work for $\text{Spin}(N)_\kappa$. We expect that the construction extends to any WZW model $G_\kappa$, leading to many new self-dualities of these models and associated non-invertible symmetries. It is possible that for other $G$, the relation between the level $\kappa$ and subgroup $\mathbb{Z}_p$ in (5.42) changes. We will return to this point in upcoming work.

An important remark is that, in general, conformal symmetry of the sigma model does not play a role in our construction; we are able to construct non-invertible symmetries in NLSMs without conformal invariance, as in the example of the nilfold in section 5.4. Of course, in the absence of conformal symmetry, the geometry of the NLSM will generically depend on the renormalisation group (RG) scale, and so the expectation is that the self-duality conditions can *a priori* only be satisfied at a particular energy. There are two possibilities in this scenario. One is to interpret this as the existence of some symmetry enhancement at some intermediate point of the RG

flow, not dissimilar to usual symmetry enhancements in the deep IR. Another is that the combinations of parameters $E_{mn}$ appearing in the self-duality conditions could turn out to be protected under the RG flow, precisely by the non-invertible symmetry, which would then be present at all energies (a similar phenomenon occurred in [76]). It would be extremely interesting to verify if this is the case.

It is more straightforward to look at conformal NLSMs, where our defects are present at all scales. Topological defect lines in 2d CFT have been heavily studied (see e.g. [21, 22, 24–36] and references therein), with most of the developments focusing on rational CFTs. To the authors' knowledge, the existence of non-invertible defects in non-rational CFTs has only recently been addressed in some examples with Calabi-Yau or K3 target spaces in [77, 78]. From our approach, rationality of the conformal NLSM or lack thereof will again not play a role. Indeed, it is easy to come up with examples of non-rational CFTs that (for some fixed parameters) will contain self-duality defects: it suffices to consider a target space where the base of the fibration $N$ is non-compact. It would be interesting to try to make contact between the approach of the present work and these recent constructions.

We believe that our work opens the door to many interesting questions. We conclude with a brief discussion of some of them.

- One important point, already mentioned in the main text, is to understand the action of these non-invertible symmetries on the operators of the NLSM. The naïve expectation, in light of studies of the compact boson as well as other examples of non-invertible symmetries, is that it will exchange order and disorder operators (which would correspond to momentum and winding for spaces with 1-cycles) and will have a non-trivial kernel, which is to say that it will annihilate some part of the states. Once the action of the symmetry is known, it should be possible to use the symmetry to derive new Ward identities (similarly to e.g. [18]). If our expectations are correct, these would be constraints relating correlation functions involving order and disorder operators. WZW models seem to be a promising playground for these investigations. Hopefully, this analysis would also shed light on the target space interpretation of the non-invertible defect, and the possible relation with T-folds [46, 64, 75], where the transition functions between patches can exchange the two conserved currents associated with the dual isometries.

- A technical point in the construction of self-duality defects is the boundary conditions that one imposes on the gauge fields. Following [14], we have set $c|_\gamma = 0$ (1.1), which is a sufficient condition that guarantees that the defect will be topological. However, as we have already noted in section 4.4 (around equation (4.38)), weaker conditions can be set: only the pull-back of $c$ to $\gamma$ makes an appearance, and so it seems to be enough to set the tangential components of $c$ to zero on $\gamma$, with the normal ones unconstrained. This leads to

the same topological theory living on $\gamma$. It would be interesting to understand if the defect built with these weaker boundary conditions has any different properties to the standard ones, in particular as far as the analysis of the equations of motion in its presence is concerned, and as a consequence regarding its action on vertex operators. It would also be very interesting to elucidate how said weaker condition would translate to lattice constructions of self-duality defects.

- NLSMs can be used to describe the dynamics of strings propagating in a given target space. Indeed, while in the present work we have focused on the 2d worldsheet picture, we have also had recourse to invoke intuition from String Theory. It is then natural to extend the half-space gauging construction to supersymmetric NLSMs. It is straightforward to gauge isometries in NLSMs with (1,1) worldsheet supersymmetry by coupling to (1,1) vector supermultiplets [79] and to then add a Lagrange multiplier term to derive T-duality. The constraints on the target space are exactly the same as for the gauging of the bosonic NLSM. This can be modified to give a $\prod_m \mathbb{Z}_{p_{(m)}} \subset U(1)^d$ gauging followed by a T-duality, which is done by introducing factors of $p_{(m)}$ in the T-duality construction, in the same way as was done for the bosonic case in this paper. The half-space construction then results in a non-invertible symmetry for the supersymmetric NLSM provided the self-duality constraints are satisfied. It would also be interesting to understand if other defects could be constructed for supersymmetric NLSMs, related to other string dualities.

- It is interesting to ask if the non-invertible symmetry of the NLSM on a given worldsheet leads to a non-invertible symmetry of String Theory. It is believed that a theory of quantum gravity should have no global symmetries [80, 81]. We want to emphasise that our construction need not be a counterexample to this conjecture. It has already been studied in some examples in [82, 83] that Ward identities related to known non-invertible symmetries in two dimensions tend to depend explicitly on the topology of the worldsheet (even if their construction does not). Therefore, even if at each order in the string coupling there appears to be a Ward identity, the sum over string loops will mean that none of them are satisfied by the full correlator. All the relevant features are present for the same phenomenon to happen for our defect, although, of course, it would also be interesting to check it explicitly. If this were not the case, it would remain an intriguing possibility that there is a non-invertible *duality* symmetry of String Theory which can be regarded as a discrete gauge symmetry.

- We have focused throughout this work on the construction of a particular defect, arising from the half-space gauging of a subgroup of the isometry symmetry. In the compact boson, many other symmetries can be constructed

by gauging different subgroups of momentum and winding. A particularly interesting setup was discussed in [39] that allowed the construction of non-invertible defects at any value of the radius. Exploring whether these types of constructions can be extended to more general NLSMs, and which self-duality conditions they should satisfy in this case, is another interesting avenue.

- A different approach to the study of T-duality symmetries was used in [38, 41], that consists on finding $O(d, d, \mathbb{Z})$ transformations of the generalised metric that can leave the theory on both sides of the defect invariant, and then identifying which particular combination of gaugings of momentum and winding they correspond to. It would also be interesting to explore this method in more general NLSMs; which may require the formalism of Double Field Theory [64, 84] in order to deal with topologically non-trivial $b$-field configurations.

- With the objective of identifying all topological defects and their properties in a theory-agnostic manner, it would be interesting to identify the Symmetry Topological Field Theory (SymTFT) [85] which encodes the defects constructed in this work. This is a topological theory in one higher dimension which encodes all the topological operators present in a given QFT living on one of its boundaries. In this direction, WZW models again seem to be a promising case study, as there is a well-developed story of how they can be realised on the boundary of Chern-Simons theories [86]. In order to include the self-duality defects, and not only the Verlinde lines, it is likely this would require a discrete gauging of the 3d theory, along the lines of [87].

- Finally, it is one of the fundamental goals in the life of a symmetry to help constrain the possible behaviours of complicated physical systems [1]. This is also true for generalised symmetries, the consequences of which have been studied in two dimensions, e.g. in [76, 88–92]. It is both interesting and important to understand the physical implications of the non-invertible symmetries constructed in this work.

## Acknowledgments

It is a pleasure to thank Jeremias Aguilera Damia, Riccardo Argurio, Stephanie Baines, Lakshya Bhardwaj, Michele Del Zotto, Giovanni Galati, Iñaki García Etxebarría, Shota Komatsu, Julio Parra-Martínez, Diego Rodríguez-Gómez and Ignacio Ruiz for interesting discussions. GAT and CMH are supported by the STFC Consolidated Grants ST/T000791/1 and ST/X000575/1. CMH was also partially supported by the Swedish Research Council under the grant no. 2021-06594 while the author was in residence at the Mittag-Leffler Institute in Djursholm, Sweden, during Febru-

ary and March 2025. MVCH is supported by a President's Scholarship from Imperial College London.

## A  Differential geometry glossary

In this short appendix we summarise the notation and conventions that we make use of throughout the text. Firstly, the normalisation for the components of a differential $p$-form $\omega$ is

$$\omega = \frac{1}{p}\omega_{i_1\ldots i_p}\mathrm{d}X^{i_1}\wedge\cdots\wedge\mathrm{d}X^{i_p}\,. \tag{A.1}$$

This slightly non-standard convention matches that of [46], which several aspects of our construction are based on. In particular, this implies that the contraction between the 3-form $H$ and the Killing vector $k_m$ can be written

$$\iota_m H \equiv \iota_{k_m} H = k_m^i H_{ijk}\,\mathrm{d}X^j\wedge\mathrm{d}X^k\,. \tag{A.2}$$

We use a straight d to denote the exterior derivative on target space (so, e.g., $H = \mathrm{d}b$), and a slanted $d$ to denote the worldsheet exterior derivative (e.g. in (3.2)). The two are related by the pull-back of differential forms along $\Phi : W \to M$.

The setup of our construction requires a target space $M$ with a set of $d$ compact abelian isometries, generating an isometry group $\mathrm{U}(1)^d$. The Killing vectors are denoted $k_m$. As described in section 3.1, provided that the isometries act freely and properly discontinuously, the orbit space $N = M/\mathrm{U}(1)^d$ is a manifold and $M$ is a $\mathrm{U}(1)^d$ bundle over $N$. A set of coordinates on $N$ are denoted by $Y^\mu$ and the fibre coordinates are denoted by $X^m$. These can be chosen such that the Killing vectors are as in (3.8).

The bundle geometry can be described in terms of quantities on the base (e.g. a metric, connections, and their curvatures). Following [46], we refer to such objects as *basic*. A form $\omega$ is basic if it does not depend on the fibre coordinates $X^m$ and has no component along the fibre directions. In coordinate-independent language, these requirements are

$$\mathcal{L}_m\omega = 0, \qquad \iota_m\omega = 0 \tag{A.3}$$

respectively. A form satisfying the former is said to be *invariant*, while a form satisfying the latter is *horizontal*. Note that, for example, closed horizontal forms are automatically basic.

Similarly, a metric $\bar{g}$ is horizontal if

$$\bar{g}(k_m, V) = 0\,, \tag{A.4}$$

for all vector fields $V$, and for all the Killing vectors $k_m$. In the coordinates described above, this implies that the only non-zero components of $\bar{g}$ are $\bar{g}_{\mu\nu}$. Furthermore, if

the $k_m$ are Killing vectors of $\bar{g}$ (i.e. $\mathcal{L}_m \bar{g} = 0$) then the metric is called invariant, so it does not depend on the $X^m$ coordinates. A metric which is horizontal and invariant is then basic.

In the text, the connections $A^m$, $\widetilde{A}_m$, their curvatures $F^m$, $\widetilde{F}_m$, and the metric $\bar{g}$ are the basic quantities which ultimately determine the target space geometry and its behaviour under discrete gauging (including T-duality).

# B  Gauging the Wess-Zumino term on a worldsheet with boundary

In this appendix, we review the construction introduced in [61] to couple a background gauge field for the isometry symmetry in a WZ action on a worldsheet with boundary. As we will see, it is not enough to simply promote derivatives to covariant derivatives, and one should use Noether's method instead; this also allows us to identify the conditions for 't Hooft anomaly cancellation, analogously to the discussion in section 3.1.

The setup is the one described in section 4.3 and summarized in Figure 3. The ungauged action is as in (4.28), although here we drop the metric term since it can be gauged by standard minimal coupling as in (3.30) and is unaffected by the boundary,

$$S_{WZ} = \frac{1}{3} \int_{V_+} H_{ijk}(X) dX^i \wedge dX^j \wedge dX^k - \frac{1}{2} \int_{\Delta} \beta_{ij}(X) dX^i \wedge dX^j \equiv S_H - S_\beta. \quad \text{(B.1)}$$

The isometry symmetry acts as

$$X^i \to X^i + \alpha^m k_m^i(X). \quad \text{(B.2)}$$

Noether's method proceeds by considering the parameters to be local, $\alpha^m = \alpha^m(\sigma)$, in order to find the associated conserved current and elucidate the required couplings to the gauge fields $c^m$. We begin by looking at the variation of the first term in (B.1). At first order in $\alpha$,

$$\begin{aligned}
\delta S_H &= \int_{V_+} H_{ijk} \, d\alpha^m k_m^i \wedge dX^j \wedge dX^k \\
&\quad + \int_{V_+} \alpha^m \underbrace{\left( H_{ijk} \partial_l k_m^i dX^l + \frac{1}{3} k_m^l \partial_l H_{ijk} \, dX^i \right)}_{\propto \mathcal{L}_m H = 0} \wedge dX^j \wedge dX^k + O(\alpha^2) \quad \text{(B.3)} \\
&= \int_{V_+} (\iota_m H)_{jk} \, d\alpha^m \wedge dX^j \wedge dX^k + O(\alpha^2), \quad \text{(B.4)}
\end{aligned}$$

where we have used (3.3), which imposes that $\mathcal{L}_m H = 0$ in order for (B.2) to be a global symmetry in the first place. Condition (3.15) now implies

$$\delta S_H = \frac{1}{2\pi} \int_{V_+} d\Phi_+^*(v_m) \wedge d\alpha^m = \frac{1}{2\pi} \int_{\Gamma_+} \Phi_+^*(v_m) \wedge d\alpha^m + \frac{1}{2\pi} \int_{\Delta} \Phi_+^*(v_m) \wedge d\alpha^m + O(\alpha^2). \quad \text{(B.5)}$$

Next we compute the variation of the second term in (B.1), $S_\beta$,

$$\delta S_\beta = \int_\Delta \beta_{ij}\, d\alpha^m k_m^i \wedge dX^j + \int_\Delta \alpha^m \underbrace{\left(\beta_{ij}\partial_k k_m^i dX^k + \frac{1}{2}k_m^l \partial_l \beta_{ij} dX^i\right)}_{\propto \mathcal{L}_m \beta = 0} \wedge dX^j + O(\alpha^2),$$

(B.6)

where we have used (4.32), which is a condition for (B.2) to be a global symmetry of the WZ action in a worldsheet with boundary. We are left with

$$\delta S_\beta = -\int_\Delta \Phi_+^*(\iota_m\beta) \wedge d\alpha^m + O(\alpha^2).$$

(B.7)

In total, we find

$$\delta S_{WZ} = \delta S_H - \delta S_\beta = \frac{1}{2\pi}\int_{\Gamma_+} \Phi_+^*(v_m) \wedge d\alpha^m + \int_\Delta \Phi_+^*\left(\iota_m\beta + \frac{1}{2\pi}v_m\right) \wedge d\alpha^m + O(\alpha^2).$$

(B.8)

In order for (B.2) to be a global symmetry, we must demand that the variation vanishes when the $\alpha^m$ are constants. Naïvely this appears to be the case, however the second integral in (B.8) is on $\Delta$, which is an auxiliary surface. The theory defined on $\Gamma_+$ will only have the global symmetry (B.2) if the variation vanishes when this second term is written on $\gamma = -\partial\Delta$, which is part of the worldsheet. Using Stokes theorem, this is the case if we impose that

$$\left.\iota_m\beta + \frac{1}{2\pi}v_m\right|_Y = \frac{1}{2\pi}dh_m.$$

(B.9)

This is the constraint given in (4.31) for (B.2) to be a global symmetry of the NLSM on a worldsheet with boundary.

Let us now discuss the gauging of this global symmetry. Using (B.9), for arbitrary $\alpha^m(\sigma)$ we have

$$\delta S_{WZ} = \frac{1}{2\pi}\int_{\Gamma_+} \Phi_+^*(v_m) \wedge d\alpha^m - \frac{1}{2\pi}\int_\gamma \Phi_+^*(h_m)\, d\alpha^m + \mathcal{O}(\alpha^2).$$

(B.10)

These $\mathcal{O}(\alpha)$ variations can be cancelled by coupling gauge fields $c^m$ that transform as $c^m \to c^m + d\alpha^m$ via

$$S_{\text{gauged } WZ} = S_{WZ} - \frac{1}{2\pi}\int_{\Gamma_+} \Phi_+^*(v_m) \wedge c^m + \frac{1}{2\pi}\int_\gamma \Phi_+^*(h_m)c^m + \mathcal{O}(c^2).$$

(B.11)

This leads to the $\mathcal{O}(c)$ contributions in equation (4.37). One then has to study $\mathcal{O}(\alpha^2)$ variations to the action and the $\mathcal{O}(c^2)$ couplings required to gauge them. There are further constraints for this gauging to be possible, otherwise there is a 't Hooft anomaly. The gauging is possible if the freedom to shift $v_m$ by a closed 1-form and $h_m$ by a constant function can be used to set [47, 61]

$$\mathcal{L}_m v_n = 0, \quad \iota_m v_n + \iota_n v_m = 0, \quad \mathcal{L}_m h_n = 0.$$

(B.12)

For the case where the worldsheet has no boundary, introducing $h_m$ is not required and the first two constraints are those of (3.11). When the worldsheet has a boundary, the final constraint is also necessary (4.36).

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
