# Peer review of "Non-invertible symmetries of two-dimensional Non-Linear Sigma Models"

_SciPost Physics_

## Round 2 · Referee Report · Anonymous (Referee 1) · 2025-7-17

Strengths

The paper is well written and clear. It extends previous results regarding T-duality defects to other 2d QFTs.

Weaknesses

-

Report

This paper provides a clear an interesting extension of the construction of T-duality defects, extensively discussed in the case of the 2d compact boson, to the case of more generic non linear sigma models. The construction still relies on the method of half-space gauging which sends the theory to its dual version upon a topological manipulation. The paper is clear and nicely written;  I therefore recommend it for publication, though I would like the authors to answer/comment to the following: 1. At the end of pag. 4, the authors comment on the fact that, in order to perform the half space gauging, the T-duality defect must be placed on homologically trivial cycles. Even if I agree that one cannot perform a gauging on half space when the cycle is non-trivial, the statement, as written, is a little bit confusing since the T-duality defect can be placed on any cycle of the world sheet (contractible or not). See e.g. reference [36] for explicit constructions of partition function of the compact boson on a torus with the duality defect placed on non-trivial cycles. 2. Regarding the last comment in Section 2.2 on page 9, I would like to point out that, by coupling to the flat gauge field, one can obtain either of the Lagrangians with radius R/p or p/(2\pi R) depending on which gauge field is integrated out first. 3. Should the last term in eq. 4.20 be \Phi^_+ instead of \Phi^_- ? 4. In the second paragraph on page 59, it is claimed that on a worldsheet without non-trivial 1-cycles, the T-duality defect is trivial, it reduces to T-duality and is invertible. I do not agree with this statement. Even though in this case the flat gauge field is trivial in empty space, one can still consider inserting this defect together with other local operators in the theory. In such cases, one will detect the non-invertibility of the defect, as it will project out operators charged under the symmetry being gauged. 5. At the beginning of page 52, the authors attempt to interpret the condition for the existence of the duality defect in non-conformal theories as a symmetry enhancement at some energy scale. I do not fully understand this argument. All derivations in the paper are carried out using the Lagrangian description of the sigma model. Since these theories are asymptotically free in two dimensions, I would expect that, if the parameters of the Lagrangian are fixed as discussed in Section 4.5, the theory exhibits the non-invertible symmetry already in the UV, making it an exact symmetry of the model. As a side comment, I believe one could verify this by computing the one-loop beta function for the coupling and showing that quantum corrections do not spoil the duality conditions. 6. One curiosity I would like to ask is: do 2d SCFTs related by mirror symmetry fit into this framework? Specifically, can 2d mirror symmetry be used to construct duality defects in the same spirit as those discussed in this paper?

Requested changes

-

Recommendation

Ask for minor revision

---

## Round 2 · Referee Report · Anonymous (Referee 2) · 2025-9-26

Report

The article concerns the presence of non-invertible duality symmetries in (1+1)d nonlinear sigma models.
The authors show that, given specific choices of the radius and WZ level, nonlinear sigma models with a global U(1) isometry can host interesting duality-type symmetries, which in general are not Verlinde lines.
The paper is very well written and the results clear and crisp. The authors have already published interesting follow-ups and I expect several other applications to appear in the future.
I thus happily recommend this paper for publication in SciPost.

Recommendation

Publish (easily meets expectations and criteria for this Journal; among top 50%)

---

## Editorial Decision

resubmitted